

# Investigating the sensitivity of soil respiration to recent snow cover changes in Alaska using a satellite-based permafrost carbon model

Yonghong Yi[1,2*], John S. Kimball[3], Jennifer D. Watts[4], Susan M. Natali[4], Donatella Zona[5,6], Junjie Liu[1], Masahito Ueyama[7], Hideki Kobayashi[8], Walter Oechel[5,9], Charles E. Miller[1]

[1]Jet Propulsion Laboratory, California Institute of Technology, Pasadena, CA, USA.
[2]Joint Institute for Regional Earth System Science and Engineering, University of California, Los Angeles, CA, USA.
[3]Numerical Terradynamic Simulation Group, The University of Montana, USA
[4]Woods Hole Research Center, Falmouth, MA, USA.
[5]Department of Biology, San Diego State University, San Diego, CA, USA
[6]Department of Animal and Plant Sciences, University of Sheffield, Western Bank, Sheffield, S10 2TN, UK
[7]Graduate School of Life and Environmental Sciences, Osaka Prefecture University, Sakai, Osaka 599-8531, Japan
[8]Institute of Arctic Climate and Environment Research, Research Institute for Global Change, Japan Agency for Marine-Earth Science and Technology, Yokohama, Kanagawa 236-0001, Japan
[9]Department of Geography, College of Life and Environmental Sciences, University of Exeter, Exeter, EX4 4RJ, UK

*Correspondence to*: Yonghong Yi (yonghong.yi@jpl.nasa.gov)

**Abstract.** The contribution of soil heterotrophic respiration to the boreal-Arctic carbon ($CO_2$) cycle and its potential feedback to climate change remain poorly quantified. We developed a remote sensing driven permafrost carbon model at intermediate scale (~1 km) to investigate how environmental factors affect the magnitude and seasonality of soil heterotrophic respiration in Alaska. The permafrost carbon model simulates snow and soil thermal dynamics, and accounts for vertical soil carbon transport and decomposition at depths up to 3 m below surface. Model outputs include soil temperature profiles and carbon fluxes at 1-km resolution spanning the recent satellite era (2001-2017) across Alaska. Comparisons with eddy covariance tower measurements show that the model captures the seasonality of carbon fluxes, with favorable accuracy in predicting net ecosystem $CO_2$ exchange (NEE) in both tundra (R > 0.8, RMSE = 0.34 g C m$^{-2}$ d$^{-1}$) and boreal forest (R > 0.73, RMSE = 0.51 g C m$^{-2}$ d$^{-1}$). Benchmark assessments using two regional in-situ datasets indicate that the model captures the complex influence of snow insulation on soil temperature, and the temperature sensitivity of cold-season soil respiration. Across Alaska, we find that seasonal snow cover imposes strong controls on the contribution from different soil depths to total soil carbon emissions. Earlier snow melt in spring promotes deeper soil warming and enhances the contribution of deeper soils to total soil respiration during the later growing season, thereby reducing net ecosystem carbon uptake. Early cold-season soil respiration is closely linked to the number of snow-free days after land surface freezes (R = -0.48, p < 0.1), i.e. the delay in snow onset relative to surface freeze onset. Recent trends toward earlier autumn snow onset in northern Alaska promote a longer zero-curtain period and enhanced cold-season respiration. In contrast, southwestern Alaska shows a strong reduction in the number of snow-free days after land surface freeze onset, leading to earlier soil freezing and a large reduction in cold-season soil respiration. Our results also show non-negligible influences of sub-grid variability of surface





conditions on the model simulated $CO_2$ seasonal cycle, especially during the early cold season at 10-km scale. Our results demonstrate the critical role of snow cover affecting the seasonality of soil temperature and respiration and highlight the challenges of incorporating these complex processes into future projections of boreal-Arctic carbon cycle.

## 1 Introduction

Warming in the northern high latitudes ($> 50°N$) is occurring at roughly twice the global rate, and has trigged a series of changes in boreal and Arctic ecosystems including earlier and longer growing seasons, widespread soil thawing and permafrost degradation (Jeganathan et al., 2014; Liljedahl et al., 2016), with large impacts on the regional carbon cycle (McGuire et al., 2016). Atmospheric $CO_2$ observations indicate a strong increase in the seasonal amplitude of the northern carbon cycle, which may reflect an enhancement of net carbon uptake during the growing season or soil carbon emissions during cold season in northern ecosystems (Graven et al., 2013). However, there is a lack of consensus on whether increased vegetation productivity or enhanced respiration drives such changes, due to sparse in-situ measurements, uncertainties in satellite remote sensing retrievals and model simulations (Fisher et al., 2014; Forkel et al., 2016; Parazoo et al., 2016; Wenzel et al., 2016). For example, there is a large discrepancy on the contribution of cold-season respiration to the annual carbon budget in the boreal-Arctic ecosystems (Zona et al., 2016; Euskirchen et al., 2017; Natali et al., 2019a). In addition, potential release of a large amount of carbon currently sequestered in perennially frozen soils in the northern high latitudes adds additional uncertainty in assessing the response of boreal-Arctic ecosystems to future climate change (Schuur et al., 2015).

Pronounced changes have occurred in the northern high latitudes, especially during the shoulder seasons. Satellite remote sensing datasets over the past several decades indicate reductions of 0.8-1.3 days decade[-1] in the mean duration of the frozen period in the northern high latitudes (Kim et al., 2015) and ~3-4 days decade[-1] in the snow cover duration across the Northern Hemisphere mostly due to spring snow cover reduction (Hori et al., 2017; Bormann et al., 2018). Strong warming in both spring and fall has significantly reduced snow cover during the shoulder seasons; however, there is large spatial variability across the region, partly due to more variable snow cover conditions during fall and winter (Brown and Derksen, 2013; Hori et al., 2017). Climate models project continued strong warming during the spring and fall in the Arctic, and increases in previously rare winter rain events (Bintanja and Andry, 2017). How the boreal-Arctic carbon cycle responds to such changes remains to be understood.

Previous studies reported that the combination of warming and a longer snow-free season has led to widespread greening and enhanced vegetation productivity in the northern latitudes especially during the early growing season (Aurela et al., 2004; Humphreys and Lafleur, 2011; Buermann et al., 2013; Pulliainen et al., 2017). However, a detailed understanding of how soil respiration and other belowground processes respond to climate variability, especially during the cold season, remains elusive. Surface warming and a longer snow-free season are associated with earlier soil thawing and deeper active layer thickness (ALT) in permafrost regions, which can result in enhanced soil respiration and reduced annual net carbon uptake (Lund et al., 2012; Yi et al., 2018). Moreover, ALT deepening in





permafrost regions will likely lead to a longer zero-curtain period (i.e. soil temperature persists around 0 °C) especially in the deeper active layer and can even form talik and further accelerate permafrost thawing (Connon et al., 2018; Yi et al., 2019). These changes may promote even more soil carbon losses particularly during the cold season, reinforcing a positive permafrost carbon feedback (Parazoo et al., 2018). On the other hand, the timing and magnitude of autumn snowfall determine the onset and rate of soil freeze-up, which affects soil microbial activity and soil respiration during

fall and early winter (Zona et al., 2016; Arndt et al., 2019). Better understanding of how snow cover trends are affecting soil respiration is needed to inform projections of potential response of the boreal-Arctic carbon cycle to climate change.

Landscape-level processes can affect the amount and age of soil carbon released to the atmosphere (Hobbie et al.,

2000). An important feature of boreal-Arctic landscapes is strong surface heterogeneity, driven by relatively fine scale microtopographic variability on the order of 0.1-10 meters (Zona et al., 2011; Kumar et al., 2016; Grant et al., 2017a, b), which can influence coarser landscape level behavior. However, current large-scale models generally operate at scales of 10-100 km, and are too coarse to resolve finer scale surface heterogeneity and its influence on active layer dynamics and soil carbon decomposition (Yi et al., 2015; Tao et al., 2019). Satellite or airborne remote sensing can

provide information on land surface heterogeneity across large extents, and may provide critical constraints on model predictions of regional active layer changes, soil carbon and permafrost vulnerability. Therefore, the objective of this study was to develop a process-based permafrost carbon model mainly driven by satellite remote sensing data. The model was designed at an intermediate scale (~1 km) that is efficient for regional runs but also able to bridge the gap between very fine-scale (~tens of meters) ground measurements and large-scale (~tens of kilometers) earth system

simulations. The model simulations were conducted over a multi-year period (2001-2017) across Alaska to study how soil carbon emissions and the seasonal carbon cycle are responding to recent climate and snow cover trends.

## 2 Methods

### 2.1 Model description

The Remote Sensing driven Permafrost Model (RS-PM) developed in Yi et al (2018; 2019), was coupled with a

terrestrial carbon flux (TCF) model (Yi et al., 2015) to investigate the climate sensitivity of carbon fluxes, especially soil carbon emissions, across boreal-Arctic ecosystems in Alaska, and with a particular focus on the shoulder seasons.

The RS-PM follows the prototype of a detailed permafrost hydrology model (Rawlins et al., 2013; Yi et al., 2015), but has a flexible structure designed to use satellite remote sensing data as key model drivers and for model

parameterization. The RS-PM uses a numerical approach to simulate soil freeze/thaw (F/T) and temperature profiles down to 60 m below the surface, using 23 soil layers with increasing layer thickness at depth. The first 10 soil layers cover the top 1-m, with layer depths of 0.01, 0.03, 0.08, 0.13, 0.23, 0.33, 0.45, 0.55, 0.70 and 1.05 m. The model also accounts for the effects of seasonal snow cover evolution, organic soil and soil-water phase change on soil F/T processes. Satellite-based land surface temperature (LST) and snow cover time series data were used as model drivers.





Soil thermal properties were parameterized using soil moisture data from the Soil Moisture Active Passive (SMAP) Level 4 (L4) data assimilation system (Reichle et al., 2017). RS-PM validation using in-situ measurements shows favorable model accuracy for ALT (mean R = 0.60, bias = 1.58 cm, RMSE = 20.32 cm) and zero-curtain period (mean R=0.60, RSME=19 days) simulations, especially for North Slope Alaska (Yi et al., 2018, 2019).

We coupled the RS-PM and TCF models to represent the influence of permafrost active layer processes on net ecosystem $CO_2$ exchange (NEE) and its component carbon fluxes. The TCF model uses a light use efficiency (LUE) algorithm driven by satellite FPAR (Fraction of vegetation canopy absorbed Photosynthetically Active Radiation) estimates to calculate vegetation productivity and litterfall inputs to a soil decomposition model:

$$GPP = \varepsilon \times FPAR \times PAR \qquad (1)$$

$$\varepsilon = \varepsilon_{max} \times T_{mn\_scalar} \times SM_{mn\_scalar} \qquad (2)$$

where GPP is the gross primary productivity (g C m$^{-2}$ d$^{-1}$), $\varepsilon$ (g C MJ$^{-1}$) is the LUE coefficient converting canopy absorbed photosynthetically active solar radiation (APAR; MJ m$^{-2}$ d$^{-1}$) to biomass. The biome-specific maximum LUE coefficient ($\varepsilon_{max}$) was reduced for suboptimal temperature and moisture conditions represented by the scalars $T_{mn\_scalar}$ and $SM_{mn\_scalar}$ to estimate $\varepsilon$. MODIS nighttime LST and SMAP L4 rootzone (0-1 m depth) soil moisture

records were used to estimate these rate scalars using a simple liner ramp functions (Yi et al., 2015). Vegetation net primary productivity (NPP) was estimated as a fixed portion of GPP for each biome type based on an assumption of conservatism in vegetation carbon use efficiency within similar plant functional types. Annual litterfall was assumed to be equal to annual NPP without accounting for the impact of disturbance events.

Our soil decomposition model uses multiple litter and soil organic carbon (SOC) pools to characterize the progressive decomposition of fresh litter to more recalcitrant materials, which include three litterfall pools, three SOC pools with relatively fast turnover rates, and a deep SOC pool with slow turnover rates (Thornton et al., 2002). The litterfall carbon inputs were first allocated to the three litterfall pools component and then transferred to the SOC pools through progressive decomposition. In the previous study (Yi et al., 2015), the litterfall and SOC pools were arbitrarily

distributed at different soil depths within the top 3 m soils to account for depth-dependent differences in litterfall and soil organic matter substrate quality. However, in this study, we model the profile of the carbon pools through introducing a vertical dimension $z$ and accounting for the vertical carbon transport across the z dimension (Elzein and Balesdent, 1995; Koven et al., 2013a):

$$\frac{\partial C_i(z)}{\partial t} = R_i(z) + \sum_{j\neq i}^{n}(1 - r_j)T_{ji}k_j(z)C_j(z) + \frac{\partial}{\partial z}\left(D(z)\frac{\partial C_i}{\partial z}\right) - k_i C_i(z) \qquad (3)$$

where $R_i$ (g C m$^{-3}$ d$^{-1}$) is the carbon input from litterfall allocated to pool $i$ through the profile, and $T_{ji}$ is the fraction of carbon directed from pool $j$ to pool $i$ with fraction $r_j$ lost as respiration; $k_i(k_j)$ is the decomposition rate (day$^{-1}$) of carbon pool $i$ $(j)$, which was derived as the product of a theoretical maximum rate constant and dimensionless multipliers for soil temperature and liquid water content constraints to decomposition, using the RS-PM model outputs. The diffusivity $D$ was used to account for vertical diffusive soil C transport while vertical C transport due to

advection was ignored here. Constant D values were assigned to permafrost (5.0 cm$^2$ yr$^{-1}$) and non-permafrost (2.0





cm$^2$ yr$^{-1}$) regions within the top 1m soil, and then linearly decreased to 0 at the 3 m below surface (Koven et al., 2013a). The boundary conditions at the soil surface were defined as:

$$D\frac{\partial c_i}{\partial z} = R_{s,i} \text{ at z=0} \tag{4}$$

where $R_{s,i}$ is the carbon input rate (g C m$^{-3}$ d$^{-1}$) to the three surface litterfall pools. A zero-flux was assigned at the bottom of the soil carbon pool, which was set as 3 m.

## 2.2 Model inputs and parameterization

The main RS-PM inputs include LST, snow cover properties and soil moisture from global satellite and reanalysis data products. LST and soil moisture records from the MODIS 8-day composite dataset (MOD11A2; Wan and Hulley, 2015) and SMAP L4 9-km daily surface (5cm depth) and root zone (0-1 m depth) products (L4SM, Reichle et al., 2017) were used to define the model boundary conditions and parameterize soil thermal properties (Yi et al., 2018). MODIS 500-m snow cover extent (SCE) data (MOD10A2; Hall and Riggs, 2016) were used to downscale snow depth and density data from the MERRA2 (~0.5°) global reanalysis (Gelaro et al. 2017) to characterize sub-grid variability in snow distribution as described in Yi et al. (2019). The RS-PM model outputs include soil temperature and liquid water fraction within the soil profile, which are the major inputs to the soil decomposition model. Other primary inputs to the TCF model include MODIS normalized difference vegetation index (NDVI), nighttime LST, and MERRA2 downward solar radiation data. The NDVI data was used to estimate FPAR using a biome-specific empirical relationship (Yi et al., 2015). The nighttime LST and SMAP L4 rootzone soil moisture were used to estimate the environmental constraints on LUE and GPP. All model input datasets were reprojected into a 1km resolution Albers projection and resampled to an 8-day time step consistent with the model simulations.

Other ancillary datasets included the 30-m national land cover database (NLCD) 2011 (Jin et al., 2013), 50-m SOC estimates for Alaska (to 1-m depth; Mishra et al., 2017), and the global 9-km mineral soil texture data developed for the SMAP L4SM algorithm (De Lannoy et al., 2014). The dominant NLCD land cover type within each 1 km pixel was used to define the modeling domain, with open water and perennial ice and snow areas excluded (Fig. 1). The SOC inventory data was used to define the organic fraction of the top 10 model soil layers (~1.05 m depth), which was used to adjust the soil properties of each soil layer based on the weighted mineral and organic soil components. More details on the data processing and soil parameterization can be found in Yi et al. (2018, 2019).

A dynamic litterfall allocation scheme based on the satellite NDVI time series was used in Yi et al (2015) to account for litterfall seasonality. We revised this scheme for the present study to incorporate a vertical distribution of root turnover, required by the soil decomposition model. The total litterfall was partitioned into aboveground (leaves and woody components) and belowground (mostly fine roots) litterfall using prescribed ratios for each biome type (Table S1). A constant turnover rate for each 8-day composite period was assigned to the woody components of litterfall. The turnover rates of the other components of litterfall, i.e. leaves and fine roots, were calculated based on the annual time series of MODIS NDVI, with more litterfall generally allocated during the latter half of the year. The belowground litterfall was distributed through the rooting depth based on a vertical root distribution profile (Jackson





et al., 1996). The maximum root depth in permafrost areas was limited to the maximum thaw depth. Then, the total litterfall at each depth was first allocated to the three litterfall pools according to the substrate quality of each litterfall component, i.e., labile, cellulose and lignin fractions, and then transferred to the SOC pools through progressive

decomposition. Table S1 provides the main parameters of the TCF model for each biome type, which were largely consistent with the prior study (Yi et al., 2015).

**2.3 In-situ data and model validation**

We used four Alaska eddy covariance (EC) tower sites having multi-layer soil temperature or moisture measurements to evaluate the simulated carbon fluxes and temperature sensitivity of ecosystem respiration. Table 1 lists the relevant

site characteristics. The Atqasuk site (US-Atq) is about 100 km south of Utqiaġvik on the Alaska North Slope and consists of a mixture of tussock tundra and shrubs with some sedges and sandy soils (Davidson et al., 2016; Arndt et al., 2019). The Ivotuk site (US-Ivo) is about 300 km south of Utqiaġvik in the northern foothills of the Brooks Range and characterized as a mixed tussock tundra/moss composition on a gentle slope (Arndt et al., 2019). Soil temperature measurements were available at 5, 15, 30 cm at US-Atq and 5, 15, 30, and 40 cm at US-Ivo, with full annual cycles

recorded in 2014 and 2015. The two boreal forests sites (US-Prr and US-Uaf) are located near Fairbanks, Alaska and dominated by mature black spruce forest (Ueyama et al., 2014; Ikawa et al., 2015). The leaf area index is ~ 0.73 at US-Prr and 1.9 at US-Uaf. Understory vegetation is dominated by peat moss and feather moss. The US-Uaf is located in ice-rich permafrost, and the soil is silt-loam overlain by a 25-45 cm organic layer. Measurement records longer than 7 years were available at both forest sites; however, soil temperature measurements at the two sites show some drift

throughout the period, while soil moisture measurements are more consistent. Therefore, for the boreal sites, we used the relationship between ecosystem response and the zero-curtain period calculated from the soil moisture measurements to evaluate the temperature response of cold-season respiration. The zero-curtain period was defined as the difference between surface freeze-up and the soil freeze-up dates, where soil freeze-up was defined as the date when soil moisture dropped below 15%–20% of the annual amplitude after surface freeze-up (Yi et al., 2019).


We used two regional datasets to evaluate the model performance during the cold-season. Daily snow depth and soil temperature measurements at SNOTEL (SNOwpack TELemetry) sites across Alaska (http://www.wcc.nrcs.usda.gov) were used to evaluate the model skill in representing snow insulation effects during the cold-season, using a snow and heat transfer metric (SHTM) defined in Slater et al. (2017), which was based on the deviation of a model simulated

snow insulation curve from observations. The snow insulation curve can be characterized as an exponential relationship of attenuated soil temperature amplitude with increasing snow depth, with snow insulation influences diminishing beyond a certain depth:

$$A_{norm} = P + Q(1 - e^{-(S_{depth,eff}/R)}) \tag{5}$$

where $A_{norm}$ is the normalized temperature amplitude difference between air temperature and soil temperature,

ranging from 0 to 1. The effective snow depth $S_{depth,eff}$ describes the snow insulation impact and is the integrated monthly snow depth from October to March weighted by its duration. P and Q are empirical parameters and R is the effective damping soil depth, which can be determined using a data fitting method. We chose to evaluate the modeled



snow insulation effects using the SHTM metric, rather than directly compare the modeled and observed soil temperatures. This approach minimizes the influence of potentially large differences between the relatively coarse

(~1-km resolution) model input data and the local site observations, particularly for SNOTEL sites located in mountainous terrain.

We used the Natali et al. (2019b) in-situ winter soil flux dataset to evaluate our simulated temperature sensitivity of cold-season respiration. The $CO_2$ flux measurements were collected from previous studies using a variety of methods

(e.g. chamber, EC tower), and reported at monthly level or seasonal level when monthly data were not available as the daily average over the interval. These data represent $CO_2$ emissions from belowground ecosystems and include autotrophic (from roots) and heterotrophic respiration. Soil temperature measurements were also provided in the dataset, at varying depths. Soil temperature data at 10 cm depth were collected if available; otherwise, surface soil temperature reported in the studies were collected. The data set contains 366 data records at tundra sites and 174 data

records at boreal forest sites across Alaska from October to April during the study period (2001-2017). However, most of the data records were collected from the same sites, with 17 tundra sites and 16 boreal forest sites in total (Fig. 1). For the tundra sites, modeled ecosystem respiration and NEE from October to April are quite similar due to negligible GPP. For the boreal sites, simulated NEE can be very small or even negative (net sink) when soil temperatures approach 0 °C. We chose simulated ecosystem respiration and soil temperature values at the center of layer 3 (~8 cm)

as a representative depth and aggregated these model outputs to monthly or seasonal averages for comparison with the observation dataset.

For all of the site comparisons, the model was run using the 1-km spatial input datasets described in Section 2.2, and the model outputs at the 1-km grid cell encompassing each validation site were extracted. For the winter flux

comparison, 1-km grid cells having biome types inconsistent with the local in-situ sites were removed prior to the comparison.

## 2.4 Model analysis

The permafrost carbon model was run at 1-km resolution and 8-day time step from 2001 to 2017. The model domain encompassed the majority of the Alaska land area (~1.26 million km$^2$). The model was initialized using a two-step

spin up process prior to the transient simulations. The model was first spun-up using satellite-based LST, snow depth, and soil moisture data for 50 years to bring soil temperatures in the top ~3 m into dynamic equilibrium. The model was then run using the same meteorology inputs, simulated soil temperature and liquid water content fields over several thousand years to bring the soil carbon pools (0-3 m) into equilibrium. Due to an incomplete MODIS record in year 2000, year 2001 was used for the spin up period. The permafrost mode simulation is sensitive to the choice of

spin up year. However, our analysis focused on the interannual variability in the model simulations, and the associated model sensitivity to environmental factors, which were less affected by the choice of spin up year. In order to examine the impact of model resolution on the simulated ecosystem carbon fluxes, another set of model simulation was conducted at 10-km resolution, and the statistical distribution of the model simulated carbon fluxes was compared





between the two simulations. For the 10-km runs, all model input meteorology datasets were aggregated to the coarser resolution, and the dominant land cover type within each 10-km grid cell was used.

Correlation analysis was used to examine the sensitivity of soil freeze-up and carbon fluxes to snow cover changes and other environmental variables across Alaska. We first calculated the onset of land-surface freeze based on the MODIS LST data, which was defined as the center date of the 8-day period at which the mean LST during three

consecutive 8-day periods dropped below 0 °C. Soil freeze onset for each soil layer was then determined when the simulated soil temperature dropped below −0.35 °C and after land surface freezes; this temperature threshold corresponds to ~15-20% liquid water content in the model simulations at an Arctic Alaska site (Yi et al., 2019). The soil freeze delay at each layer was defined as the duration between land-surface freeze onset and freeze onset of the given soil layer. In permafrost areas, this was also the duration of zero-curtain period. Unfrozen conditions at depths

of 20-30 cm may persist well into the cold season and even into January; therefore, we only calculated the soil freeze onset and delay from 2001 to 2016. The number of snow-free days after land surface temperature drops below 0 °C will affect how fast and deep soil freezes (Bjerke et al., 2015). Therefore, we calculated the number of snow-free days after land surface freeze onset (defined as the difference between the snow onset and land surface freeze onset), and analyzed its correlations with the above soil freeze indices. The timing of snow onset after the summer snow-free

period was defined as the center date of the 8-day composite period when both the snow depth at this period, and the mean snow depth within the 24-day moving window was greater than 5 cm. The resulting snow onset dates were found to be very close to alternative onset dates derived from the MODIS SCE data record.

### 3. Results

#### 3.1 Model validation

Previous studies have evaluated the performance of the RS-PM model in reproducing regional ALT patterns over the Alaska domain (Yi et al., 2018) and the zero-curtain period in Arctic Alaska (Yi et al., 2019). Here we will focus on assessing the model capability in representing snow insulation effects and ecosystem carbon fluxes, particularly during the cold season.

#### 3.1.1 Model representation of snow insulation effects

The relationship between the normalized temperature amplitude difference between surface air and 20 cm depth soil conditions ($A_{norm}$), and the effective mean snow depth ($S_{depth,eff}$) derived from the Snotel observations and model simulations is shown in Fig. 2. Both the model simulations and in-situ data indicate an increase in the snow insulation effects with increasing snow depth until $S_{depth,eff}$ reaches approximately 0.3 m; this relationship is also significantly (p < 0.1) correlated with the fitted curve derived from Slater et al. (2017) (observations: R = 0.56; Model: R = 0.48).

Using an interval of 0.01 m for $S_{depth,eff}$ below 0.3 m, the RS-PM model's snow and heat metric was 0.85, indicating good performance. Similar performance was found using 5 cm depth soil temperatures. However, relatively few data points were available with $S_{depth,eff}$ lower than 0.2 m; the biases toward deeper $S_{depth,eff}$ conditions were attributed





to the model snow depth inputs. MERRA2 generally shows earlier snow accumulation compared with the MODIS

SCE data, which can lead to model overestimation of $S_{depth,eff}$. During the downscaling process, the snow depth

during the early period was reassigned as 0 when the MODIS SCE record indicated "no snow" conditions, which may

also contribute to a higher value of $S_{depth,eff}$. This highlights the challenges in generating regional fine-resolution

snow data from coarser-resolution datasets.

### 3.1.2 Model simulated carbon fluxes and temperature sensitivity

The model simulations showed overall favorable agreement with tower-based 8-day composite carbon fluxes at the

two tundra sites (Fig. 3), including strong correlation (R > 0.8, p < 0.1), minimal mean bias (0.065 g C m$^{-2}$ d$^{-1}$ for US-

Ivo and -0.015 g C m$^{-2}$ d$^{-1}$ for US-Atq), and low RMSE (0.39 g C m$^{-2}$ d$^{-1}$ for US-Atq and 0.29 g C m$^{-2}$ d$^{-1}$ for US-Ivo)

differences. However, the model showed an apparent overestimation of GPP at the US-Ivo site (bias = 0.18 gC m$^{-2}$ d$^{-1}$

$^{1}$, RMSE = 0.71 g C m$^{-2}$ d$^{-1}$). Here the aggregated land cover map indicated shrub/scrub vegetation at this site, while

in-situ survey indicates a mixing of tussock sedge, dwarf shrub and moss communities (Davidson et al., 2016).

Alternative model simulations for the site using the less productive "tundra" land cover type markedly reduced the

resulting model GPP discrepancy (bias = -0.01 g C m$^{-2}$ d$^{-1}$, RMSE = 0.42 g C m$^{-2}$ d$^{-1}$). The model biome-specific

maximum LUE term can have a large impact on the estimated GPP magnitude, and can show large variability even

within the same plant functional type (Madani et al., 2014). The model simulated GPP at US-Atq showed no apparent

bias compared with the tower measurements (bias = -0.04 g C m$^{-2}$ d$^{-1}$, RMSE = 0.34 g C m$^{-2}$ d$^{-1}$). At both sites, abrupt

decreases in the model simulated GPP occur during the peak growing season, which was mainly due to imposed low

minimum temperatures and associated LUE reductions defined by the MODIS nighttime LST observations. The

largest GPP reductions during the peak season were generally caused by very low nighttime LST. Large uncertainties

and biases can exist in the MODIS nighttime LST data, since LST was estimated mostly for clear-sky conditions. The

LST uncertainty can add additional uncertainty to the model GPP estimates, and may contribute to the apparent model

underestimation of carbon sink activity during the peak growing season (Fig. 3a, c). In addition, there is also large

uncertainty imposed from the NEE partitioning method, where different methods result in large differences (up to

more than 1 g C m$^{-2}$ d$^{-1}$) in the tower-based GPP and $R_{eco}$ estimates (Fig. 3a, c). Both the model simulations and tower

observations indicate a significant non-zero carbon flux during early cold season. The model simulated $R_{eco}$ also shows

overall similar sensitivity to surface soil temperature (Tsoil) as the tower data, including a large decrease in respiration

when surface soil temperatures drop below -2 °C (Fig. 3b, d). However, the tower-based data indicate a large amount

of scatter in the Reco-Tsoil relationship for $T_{soil}$ above 0 °C, depending on the partitioning method.

The model simulated soil temperatures showed overall good correspondence with the in-situ measurements over the

soil profile (R > 0.9 and RMSE < 2 °C; Fig. S1 and Fig. S2). Both the tower-based and model simulated soil

temperature profiles show a consistent pattern of soil warming over the growing season, followed by gradual freezing

with cold season onset; however, the soil temperature of the middle and bottom active layer can stay near 0 °C through

December. The model simulated soil respiration density profile largely follows soil temperature, with respiration peaks

during mid-summer, followed by gradual diminishment with active-layer freeze-up.



The model simulated carbon fluxes were also comparable to the in-situ data at the two boreal forest sites (Fig. 4 and Fig. S3). The model showed a slight underestimation of GPP and Reco at the US-Uaf site, with respective mean bias of -0.32 and -0.34 g C m$^{-2}$ d$^{-1}$. The model showed a slightly lower positive bias in GPP and Reco at the US-Prr site, averaging 0.16 g C m$^{-2}$ d$^{-1}$ and 0.06 g C m$^{-2}$ d$^{-1}$. At both sites, no obvious bias was observed in model simulated NEE during the growing-season, despite the model assumption of dynamic-equilibrium in the estimated carbon pools at

these two mature forest sites. A much stronger decrease in ecosystem respiration when the surface soil temperature drops below 0 °C was observed at the US-Prr site relative to the US-Uaf site (Fig. S4), which may partially reflect soil temperature measurement uncertainty (Section 2.3). Significant respiration fluxes were observed at the US-Uaf site when soil temperatures were less than 0 ºC and even below -10 °C. At the US-Uaf site, the in-situ data indicate a linear increase in the total respiration fluxes with longer zero-curtain duration during this period (Fig. 4c, n = 10, R = 0.6, p

< 0.1). The model simulations for this site indicate a similar Reco relationship with the zero-curtain period, but with a much shorter estimated zero-curtain period. The apparent model discrepancy was attributed to a lower SMAP L4SM derived mean annual soil saturation level at two boreal forest sites (~ 45-50%), relative to the in-situ measurements indicating much higher saturation (> 80%) in the deep soils. The apparent L4SM inconsistency reflects the underlying coarse resolution (9 km) of the SMAP product, which was unable to distinguish the complex local soil moisture

heterogeneity involving mature black spruce forest underlain by shallow permafrost with a poorly drained and thicker organic surface layer. We were unable to conduct a similar analysis at the US-Prr site due to the relatively short measurement record for this site comparing with US-Uaf site.

The model simulated ecosystem respiration showed a broadly similar response to surface soil temperature during the

cold-season (October to April) relative to the in-situ winter flux synthesis data from the larger Alaska domain (Fig. 5). The temperature sensitivity of the winter flux shown here is generally similar to the temperature sensitivity curve at the two tundra sites (Fig. 3b, d), when ecosystem respiration should mainly consist of soil respiration. The model indicates a rapid decrease in the soil respiration as soil temperature and unfrozen water content decrease. The in-situ data collected using chambers and the diffusion method show a similar response pattern as the model; however, the

EC data show large scattering in the respiration temperature response and evidence of large winter carbon fluxes when surface soil temperatures drop below -5 °C, especially from the open-path EC measurements (Fig. S5). At the tundra sites, model simulations showed higher correlation with observations excluding the EC-open path measurements (R = 0.49), than using all available measurements (R = 0.32). The synthesis dataset does not include any Alaskan boreal forest sites using EC-closed path measurements, and all available measurements were used in the analysis. Here, the

in-situ data indicate a more consistent winter carbon flux temperature response among different measurement methods, which was moderately correlated (R = 0.44) with the model simulated carbon flux. The model estimated soil temperature was also significantly correlated with the surface soil temperature reported for both tundra sites (R = 0.59, p < 0.01) and the two boreal forest sites (R = 0.51, p < 0.01). However, the model simulated soil temperatures showed a warm bias of 1.6 °C at the tundra sites and a cold bias of 2.3 °C at the boreal forest sites.



### 3.2 Spatial pattern and temporal trends of carbon fluxes

### 3.2.1 Annual carbon fluxes

The seasonal cycle of model simulated carbon fluxes and the soil heterotrophic respiration (Rh) from different soil depths averaged across Alaska and within different permafrost regions is shown in Fig. 6. The model simulations indicate that both GPP and Rh peak in July, while soil respiration persists well into the cold season. There is a notable difference in timing of the Rh seasonal peak from different soil depths, with a longer temporal lag for deeper soil layers. Figure 6 (c) and (d) show the seasonality of the Rh fraction from different soil depths for regions with different permafrost probability based on an ancillary permafrost map (Pastick et al., 2015; Fig. S6). Southern Alaska has relatively low permafrost probability ($\leq$ 33%), where the upper (0-13 cm) soil layer contributes to a majority of Rh and with earlier seasonal onset and peak in respiration relative to deeper soil layers. The surface soil contribution to Rh gradually decreases after the seasonal peak in May as deeper soil layers progressively warm. As the surface starts to freeze in September, Rh from deeper (> 13 cm depth) soil layers provides the major contribution to total soil respiration during the cold season (October – March). Other areas in Alaska show a similar pattern but with ~ 1-month delay in the seasonal peak of the surface Rh contribution in the colder permafrost region (permafrost probability > 67%), compared with the more southern areas.

Across Alaska, annual GPP from 2001 to 2017 shows overall positive trends mostly in western and interior Alaska (Fig. 7a), with 66.8% of areas showing positive trends and 32.9% of areas showing negative trends. However, only a very small portion of the areas show significant (p<0.1) trends. The positive GPP trends are mostly explained by a longer growing season (or snow-free season) and increasing vegetation growth, indicated by the MODIS SCE and NDVI records (Fig. S7). Areas with negative GPP trends mainly occur in southern and interior Alaska. The areas in interior Alaska with negative GPP trends also show negative trends in growing-season NDVI, and are likely associated with fire-induced vegetation loss (Ju et al., 2016). Compared with GPP, Rh shows more extensive enhancement across the region, with 88.4% (11.5%) of areas showing increasing (decreasing) respiration trends (Fig. 7b). Correspondingly, areas with strong increase in ecosystem respiration but moderate or non-significant increase in GPP show decreases in net ecosystem carbon uptake (i.e. positive NEE trends), such as the North Slope and portions of southern Alaska, while much of the Alaskan interior shows increasing net carbon uptake (i.e. negative NEE trends) due to the generally stronger increase in GPP relative to respiration (Fig. 7c). Overall, approximately 63.1% (36.9%) of the Alaska domain showed decreasing (increasing) trends in net ecosystem carbon sequestration. However, only a very small portion of the land area shows significant (p < 0.1) trends, with only 6.1% (2.1%) of areas having significant positive (negative) NEE trends.

### 3.2.2 Growing-season carbon fluxes

The model simulated growing-season Rh shows overall positive trends during the study period, while the contribution of surface ($\leq$ 13 cm) soils to total Rh shows opposite trends during the snow melting period (April to May) and the summer season (June to August, Fig. 8). During the snow melting period, the Rh trend pattern is similar to GPP, while the surface soil Rh fraction shows large positive trends in western Alaska and the North Slope. The MODIS LST



record during this period shows a general surface warming trend in western and interior Alaska during April, and across the North Slope from May to June, which contributes to an advance in seasonal snow melt in those areas (Fig. S8), and surface soil warming. From June to August, the MODIS LST data show mixed trends in interior Alaska, and overall cooling trends in southern and southwestern Alaska, which contribute to the negative model GPP trends in

those areas. However, Rh still shows extensive positive trends across Alaska, likely due to increasing trends in the deep soil (> 13 cm) respiration contribution discussed below. Correspondingly, NEE shows negative trends (i.e. increasing net carbon uptake) in interior and southern Alaska from April to May, but overall positive trends (decreasing net carbon uptake) across Alaska from June to August (Fig. S9 a, b).

The timing of snow offset or surface thaw onset shows the highest correlation with the surface soil Rh fraction during the growing season, but with opposing respiration responses during the early (April-May: R = -0.55) and peak (June-August; R ≥ 0.58) growing season (Table 2). The snow offset and spring thaw onset are highly correlated as both are mainly controlled by surface temperature (Fig. S8). Changes in the contribution of surface soils to total Rh between the early and peak growing season can be explained by a slower warming rate in deeper soils. Earlier snow melt and

reduced spring snow cover can significantly increase thermal loading into the ground, with progressive warming of underlying soils. Additional correlation analysis (Fig. 9) indicates the Rh fraction from surface soils is more closely correlated with monthly LST in April and May in areas with low permafrost probability (≤ 33%), and with LST in May and June in areas with high permafrost probability (> 67%). These periods correspond to the active snow melt period in each region, with mean snow offset date of ~ DOY 136.0±14.4 in areas with low permafrost probability, and

~ DOY 148.8±8.9 in more continuous permafrost areas. Surface air temperature has a direct effect on surface soil temperature, while snow cover has a stronger influence on deeper soil temperatures (Yi et al., 2015). This condition explains the similar low correlation between summer (June-August) LST and the Rh contribution from surface soils for the same period.

### 3.2.3 Cold-season carbon fluxes

Total Rh during the early cold season from September to November shows overall positive trends from 2001 to 2017 except for portions of interior and southwestern Alaska, while the Rh contribution from surface (≤ 13 cm) soils (hereby denoted as Rh fraction) shows a similar trend pattern as total Rh (Fig. 10). The Rh trend pattern is largely explained by regional trends in the number of snow-free days after land surface freezes (i.e. snow onset –surface freeze onset) (Fig. 10c), which shows the highest correlation with both Rh and the surface soil Rh fraction (Table 3; Rh: mean R =

-0.48; Rh fraction: mean R = -0.46) among all environmental variables examined. The number of snow-free days after land surface freeze onset shows large positive trends in southwestern Alaska and portions of southern Alaska, while negative trends are mostly shown in northern Alaska. Both total Rh and Rh fraction of surface soils generally increase with delaying surface freeze onset, but decrease with delaying snow onset, though the correlation is relatively weak (Table 3). Among the monthly snow depth data, Rh and Rh fraction show highest correlation with snow depth during

early snow season (September-October), which supports a close correlation between snow accumulation and soil respiration.





The spatial pattern in the soil respiration trends during the early cold season can be explained by the temporal lag (days) between the onset of surface freezing and freezing in deeper (23 cm) soil layers, i.e. the soil freezing delay or

the duration of the zero-curtain period in areas with permafrost occurrence (Fig. 11). The model simulations show an advance of ~ 0.78 day per year (p < 0.1) in the regional-mean soil freezing delay at 23 cm averaged across Alaska from 2001 to 2016, which is mainly driven by a delay in autumn snow cover onset (Fig. S8 d-f). However, large variations in the timing and depth of autumn snow accumulation contribute to large interannual variability in the soil freezing delay (Fig. 11c). The model simulated soil freezing delay increases with soil depth, and the soil freezing delay

at different soil depths is generally highly correlated. Soil water content is one of the major factors affecting the soil freezing delay, which explains why northern and southern Alaska show a longer delay in soil freezing than relatively drier soil regions in interior Alaska, indicated from the SMAP L4 SM record (not shown). The trends of soil freezing delay at 23 cm depth are largely determined by the number of snow-free days after land surface freeze onset (regional mean R= -0.46±0.26), with ~ 72% of areas showing significant (p < 0.1) correlation. Earlier snow onset over the

Alaska North Slope corresponds to an overall longer delay in soil freezing (i.e. longer duration of zero-curtain period), consistent with a previous study (Yi et al., 2019), while southwestern Alaska shows an overall shorter soil freezing delay due to later snow onset (Fig. S8 e-f). Soil freezing delay at 23 cm was also closely related to the snow depth during early snow season from September to October (regional mean R = 0.58±0.21), with ~ 85% of areas showing significant (p < 0.1) correlation. These results highlight the importance of the timing and accumulation rates of autumn

snow cover on soil freezing and cold-season respiration.

### 3.2.4 Impact of model resolution on the $CO_2$ seasonal cycle

Comparisons of the statistical distribution of model simulated carbon fluxes at the 1-km and 10-km resolutions show an enhanced NEE seasonal amplitude from the coarser scale model simulations (Fig. 12). A larger difference in the distributions is seen at the model simulated Rh fluxes, with slightly reduced Rh flux during summer, and enhanced

Rh flux from October to November at 10-km resolution. The largest differences in the Rh fluxes occur in October and November with daily mean differences of ~ 0.1 gC m$^{-2}$ d$^{-1}$ and a total difference of 9.8 Tg C across the entire study area from October to December, or more than 20% of the multi-year mean during the same period averaged across Alaska. This is consistent with an overall reduction in the number of days between snow onset and surface freeze onset derived from the model input datasets of LST and snow depth at 10-km resolution (Fig. 12a). The statistical

distribution of the model input snow depth data at the two resolutions also shows the largest differences in October due to more variable snow cover conditions in the early snow season, which can have a large impact on subsurface soil temperatures due to stronger insulating effects of early snow accumulation (Fig. 2; Slater et al., 2017). The model simulated GPP flux during the growing season shows only limited differences (< 2%) between the two spatial scales (not shown). However, the NEE simulations at 10-km resolution show enhanced carbon uptake during the growing

season, and enhanced carbon loss during the early cold season, with ~ 14% increases in the seasonal amplitude averaged over Alaska.



## 4. Discussion

Based on the simulations of a newly developed 1-km permafrost carbon model, we highlight the important role of snow cover variability in controlling the soil heterotrophic respiration and the $CO_2$ seasonal cycle of boreal and Arctic
ecosystems in Alaska. The large differences between model simulated soil respiration during the early cold season, and the estimated NEE seasonal amplitude at different model spatial scales also highlight potential large uncertainties in regional model simulations contributed from an inadequate representation of land surface heterogeneity.

### 4.1 Environmental sensitivity of boreal-Arctic $CO_2$ seasonal cycle

Our results show that earlier snow melting associated with spring warming enhances soil heterotrophic respiration
throughout the growing season, leading to a reduction in net carbon uptake later in the growing season in Alaska (Fig. 8 and Fig. S9). Previous studies reported that earlier snow melting generally results in enhanced vegetation productivity and carbon uptake during the early growing season, consistent with our simulations, while its impact on the net ecosystem exchange during the later growing season may vary with local climate and site conditions (Aurela et al., 2004; Humphreys and Lafleur, 2011; Pulliainen et al., 2017). The variable impact of snow on the seasonal
carbon cycle can be explained by divergent responses of vegetation productivity and Rh to soil moisture and soil temperature during the later growing season. Earlier snow melting in spring can lead to depleted soil water conditions during later growing season, resulting in a decrease in vegetation productivity and weaker net ecosystem carbon sink activity, especially in the boreal region (Buermann et al., 2013; Sulla-Menashe et al., 2018). However, our simulations indicate that deeper soil warming associated with early snow melting is mainly responsible for enhanced ecosystem
carbon loss later in the growing season. Surface warming and earlier disappearance in spring snow cover are associated with a deeper thaw depth in the permafrost region (Park et al., 2016; Yi et al., 2018). Field studies have shown that deeper permafrost thawing is associated with enhanced ecosystem respiration and thus reduced carbon sink activity during the later summer (Natali et al., 2011; Lund et al., 2012; Webb et al., 2016). A few other studies also indicate that ecosystem respiration may dominate the NEE response to spring snow cover conditions and warming in Arctic
tundra; however, divergent responses have been observed in different tundra ecosystems (Humphreys and Lafleur, 2011; Parmentier et al., 2011; Lund et al., 2012; Darrouzet-Nardi et al., 2019).

Our simulations also indicate that the arrival of seasonal snow cover and the number of snow-free days after land surface freeze play a major role controlling subsurface soil freeze-up and soil respiration during the early cold season.
Earlier snow onset relative to surface freeze onset (i.e. a short snow-free period after surface freezes) keeps the soil warm, and results in a longer soil freezing delay and zero-curtain period in permafrost areas, with enhanced soil respiration during the early cold season (Fig. 10 and Fig.11). Due to strong snow insulation effects, underlying soils can remain unfrozen for a substantial period long after surface soil freezes, i.e. the zero-curtain period. Field studies have shown persistent carbon emissions during this zero-curtain period and also throughout the winter season, while
the resulting cold season soil carbon emissions may partially offset or even exceed the growing season net carbon uptake (Elberling and Brandt, 2003; Luers et al., 2014; Webb et al., 2016; Euskirchen et al., 2017). A recent study showed that Alaska ecosystems were either a carbon source or carbon neutral during the recent observational period





(2012-2014), due to large contribution of cold season carbon emissions, with larger emissions in the early cold season based on $CO_2$ flux estimates optimized using data collected from the Carbon in Arctic Reservoirs Vulnerability

Experiment (CARVE) (Commane et al., 2017). Our simulations show a much longer soil freezing delay and zero-curtain period in 2013 than the other two years for the same overlapping period (Fig. 11c), corresponding to large net $CO_2$ fluxes during the fall in 2013 across Alaska and the North Slope region as shown in Fig. 1 of Commane et al. (2017).

However, large uncertainties are associated with cold-season carbon emissions in our estimates and other studies based on either in-situ data or atmospheric inversions. An analysis using satellite and airborne $CO_2$ observations pointed out that the current sparse $CO_2$ observational network is insufficient to constrain current and future estimates of cold-season carbon emissions and annual carbon budget of Arctic ecosystems (Parazoo et al., 2016). The in-situ winter flux synthesis dataset (Natali et al., 2019b) also shows large scatter in the winter flux response to surface soil temperature,

especially using the eddy covariance method. The in-situ dataset indicated that significant carbon loss (> 0.5 gC m$^{-2}$ d$^{-1}$) can occur even when surface soil temperature drops below -5 °C (Fig. S5). This large carbon loss at very low temperatures was not reproduced by our model, which showed a rapid decrease in soil respiration when surface soil temperatures (~8 cm) drop below -2 or -3 °C. However, previous studies have highlighted the inconsistency among different measurement methods in the Arctic and uncertainties in winter flux measurements due to significant data

loss under extreme weather conditions (Goodrich et al., 2016; Webb et al., 2016).

In summary, a lack of sufficient measurements and large uncertainties in current model estimates prevent drawing a consistent conclusion on how soil respiration responds to climate variability, particularly involving snow cover conditions and their contribution to both the annual budget and seasonality of the boreal-Arctic carbon cycle (Fisher

et al., 2014; Natali et al., 2019a). Combined observations from field studies, airborne and satellite remote sensing, with ecosystem process-based and atmospheric inversion models are needed to help bridge the gap between top-down and bottom-up estimates, and to improve our understanding of the climate sensitivity of different ecosystem components and their contribution to the annual and seasonal carbon budget (Parazoo et al., 2016; Commane et al., 2017). In this study we developed and applied a satellite-based process model that provides a valuable tool to study

the boreal-Arctic carbon cycle at an intermediate landscape (1 km) scale bridging the gap between local-scale measurements and coarser-scale global earth system models.

### 4.2 Model limitations and potential improvements

An important feature of boreal-arctic landscapes is strong surface heterogeneity, which may not be well represented in current global scale models operating at an order of tens of kilometers or more (Koven et al., 2013b; Yi et al., 2015;

Tao et al., 2019). Our comparisons between the 1-km and 10-km model simulations showed non-negligible influence of landscape heterogeneity on the model simulated $CO_2$ seasonal cycle, especially during the early cold season (Fig. 12). A total difference of ~ 9.8 Tg C from October to December across Alaska was found between the two simulations. Scaled to the larger pan-Arctic region (24.95 million km$^2$), the resulting difference represents ~194 Tg C in cold-





season carbon emissions and can account for more than 10% of the estimated total winter flux for the pan-Arctic

permafrost region (Natali et al., 2019a). The resulting uncertainty partially reflects spatial heterogeneity in autumn snow cover conditions, as well as sub-grid variability in the surface energy budget (indicated by LST). The complex relationship among soil saturation, snow accumulation and soil freezing also contributes to scale dependent differences in the soil carbon emission estimates (Outcalt et al., 1990; Oechel et al., 1997; Zhang, 2005). These results highlight a non-linear response of carbon fluxes to land surface heterogeneity across different scales (≥ 1 km). Moreover, a

number of studies have shown that microtopography at the order of a few meters exerts a strong control on permafrost thaw and carbon dynamics (Kumar et al., 2016; Liljedahl et al, 2016; Grant et al., 2017 a, b), which should be addressed in future model development.

Our current and previous assessment of the permafrost soil model also identified several areas where improvements

should be made to enhance model capabilities, especially in boreal forest. Comparisons with in situ measurements indicate larger discrepancy between model ALT simulations and in-situ data in the boreal interior of Alaska characterized by a greater density of woody vegetation, overlain by discontinuous or sporadic permafrost (Yi et al., 2018). Model simulated soil temperatures also showed a larger bias at the boreal forest sites comparing with the in-situ winter flux synthesis data (Section 3.1.2). The larger apparent uncertainty may reflect poor model representation

of the vegetation canopy influence on thermal energy loading at the soil surface. Previous studies have shown that the MODIS vegetation index, leaf area index and tree cover data are sensitive to boreal forest structure and post-fire disturbance recovery (Mastepanov et al., 2013). These datasets can be used to account for the temperature difference between soil surface and the canopy skin temperature indicated by the MODIS LST data for different vegetation categories, through using simple empirical models or more sophisticated approaches derived from canopy radiative

transfer models (Paul et al., 2004; Verhoef et al., 2007; Dolschak et al., 2015). Previous studies also indicate that topography and soil conditions are the dominant factors affecting soil moisture variability at finer scales (Crow et al., 2012), which are not sufficiently represented by the coarse-resolution (~ 9 km) soil moisture observations used as model inputs for this study. For example, our model simulations indicate a much shorter zero-curtain period at an interior Alaska boreal forest site compared with the local site measurements (Fig. 4c), and also overall shorter zero-

curtain period in interior Alaska than the Alaska North Slope and southern Alaska. This pattern was closely related to the model input SMAP soil wetness data, which indicated much drier conditions in interior Alaska. A previous model sensitivity study also indicated large model ALT uncertainties due to unresolved complexity in the soil organic fraction in the boreal-Arctic region (Yi et al., 2018). Better understanding of the scaling behavior of environmental controls on soil moisture and SOC is needed to improve model representation of active layer conditions and carbon estimates

(Mishra and Riley, 2015).

Other notable uncertainties in the model estimated carbon fluxes include insufficient representation of the soil moisture migration with permafrost thaw and winter processes. Earlier spring thaw and snow melt has been linked with active layer deepening and permafrost degradation, exacerbating the soil water deficit during the later growing season,

especially in the southern boreal forest areas (Buermann et al., 2013; Park et al., 2016; Zhang et al., 2019). Using



external soil moisture inputs, the current permafrost model was not able to fully represent this phenomenon, which requires a more complete depiction of soil water, energy and carbon processes, and linkages (Walvoord and Kuryly, 2016). On the other hand, insufficient winter process representation in our model may partly explain the inconsistency between the model simulated and observation-based temperature response curve of the winter flux especially indicated

by the EC tower-based measurements (Fig. 5). For example, field studies have shown that the soil $CO_2$ flux from microbial production during fall and winter can be trapped due to the overlying snowpack or surface ice layers (Elberling and Brandt, 2003; Raz-Yaseef et al. 2017). The trapped $CO_2$ can be rapidly released during high wind conditions or the spring thaw period, resulting in strong transient flux events, which are more likely recorded in EC measurements, but not detected in closed chamber measurements (Luers et al., 2014; Webb et al., 2016). Late-season

bursts in $CO_2$ emissions were also reported during the soil freeze-in period at a high Arctic wetland site (Mastepanov et al., 2013). However, our model currently assumes all soil respiration produced by the microorganisms is released directly to the atmosphere, without the mediation of snowpack, ice and mesoscale wind and pressure conditions on $CO_2$ emissions.

**5 Conclusion**

We developed a remote sensing driven permafrost carbon model at intermediate scale (~1 km) to evaluate the sensitivity of the seasonal and annual carbon ($CO_2$) cycle, and soil respiration to snow cover changes across Alaska during the recent two decades (2001-2017). Our results indicate that earlier snow melt onset associated with spring warming enhances soil heterotrophic respiration throughout the growing season and reduces net carbon uptake during later growing season when carbon losses from enhanced deep soil respiration may offset or exceed ecosystem carbon

gains from vegetation productivity. Soil freeze-up and early cold-season soil respiration are closely linked to the number of snow-free days after land surface freezes, i.e. the delay in snow onset relative to surface freeze onset. Recent trends toward earlier autumn snow onset in northern Alaska promote a longer zero-curtain period and enhanced cold-season respiration. In contrast, southwestern Alaska shows a longer delay in autumn snow accumulation relative to surface freeze onset, leading to earlier soil freezing and a large reduction in cold-season soil respiration. Our results

also show non-negligible influences of sub-grid variability of surface conditions on the model simulated $CO_2$ seasonal cycle, especially during the early cold season at 10-km scale. These results confirm the important control of seasonal snow cover on annual and seasonal carbon exchange of boreal-Arctic ecosystems. A nonlinear response of soil respiration to snow cover changes poses significant challenges for global earth system models in accurately projecting the pan-Arctic carbon cycle response to climate change.


*Code and data availability.* The regional model simulations will be archived and distributed for public access through the NASA ABoVE archive at the NASA ORNL DAAC (https://daac.ornl.gov/). All data used in this study were obtained from free and open data repositories. We will upload the code to GitHub and make it public.

*Author contributions.* Y.Y., J.S.K. and C.E.M. initiated the study. Y.Y did the calculations and wrote the paper. Other co-authors contributed to the data, and provided feedbacks on the final version.





*Competing interests.* The authors declare no conflict of interest.

*Acknowledgements.* This study was funded by NASA Terrestrial Ecology Program as part of the Arctic Boreal Vulnerability Experiment (ABoVE). A portion of the research was carried out at the Jet Propulsion Laboratory, California Institute of Technology, under contract with NASA. © 2020. All rights reserved.

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



**Table 1** Characteristics of the eddy covariance tower sites used for model validation.

|  | US-Atq[a] | US-Ivo[a] | US-Uaf[b] | US-Prr[c] |
|---|---|---|---|---|
| Location | 70.4696° N 157.4089° W | 68.4805° N 155.7568° W | 64.8663° N 147.8555° W | 65.1237° N 147.4876° W |
| Mean Tair (°C) | -9.4 | -8.3 | -2.9 | -2.0 |
| Annual P (mm) | 93 | 304 | 263 | 275 |
| Vegetation | Tussock tundra | Tussock Tundra | Black spruce forest | Black spruce forest |
| Permafrost | Yes | Yes | Yes | Yes |
| Observation period | 2014-2016 | 2013-2016 | 2008-2017 | 2011-2016 |
| Tsoil measurement depths (cm) | 5, 15, 30 cm | 5, 15, 30, 40 cm | 10, 20, 50, 80 125 cm* | 5, 10, 20, 30, 40, 100 cm* |
| SM measurement depths (cm) | 5, 15, 30 cm | 5, 15, 30 cm | 5, 15, 25 cm | 5, 10, 20, 30, 40 cm |

*Data were not consistent throughout the observational period*
*Site references: [a]Davidson et al., 2016; Arndt et al., 2019; [b]Ueyama et al., 2014; [c]Ikawa et al., 2015*


**Table 2** Regional mean correlation coefficient between environmental variables and Rh fraction of surface (0-13 cm) soils during growing season from 2001 to 2017. Unless indicated, the variables were calculated during the same period as the Rh fraction. The thaw onset was derived from MODIS LST data, and the snow offset was derived from MERRA2 downscaled snow depth data.

| Period | Thaw onset | Snow offset | GPP | LST |
|---|---|---|---|---|
| Rh fraction (April-May) | -0.55 | -0.55 | 0.40 | 0.48 |
| Rh fraction (June-August) | 0.66 | 0.58 | -0.24 (-0.43*) | -0.26 |

*indicates GPP from April to August.*

**Table 3** Regional mean correlation coefficient between environmental variables and surface (0-13 cm) soil contribution to total Rh during early cold season (September to November). Unless indicated, the variables were calculated during the same period as the Rh fraction.

|  | GPP | LST | SNOD | Freeze onset | Snow onset | Snow onset-freeze onset |
|---|---|---|---|---|---|---|
| Total Rh | 0.33 (0.27*) | 0.15 | 0.24** | 0.36 | -0.22 | -0.48 |
| Rh fraction (0-13 cm) | 0.13 (-0.10*) | 0.01 | 0.22** | 0.33 | -0.21 | -0.46 |

*GPP from April to November; ** Snow depth (SNOD) from September to October shows strongest correlation with the Rh and Rh fraction.*





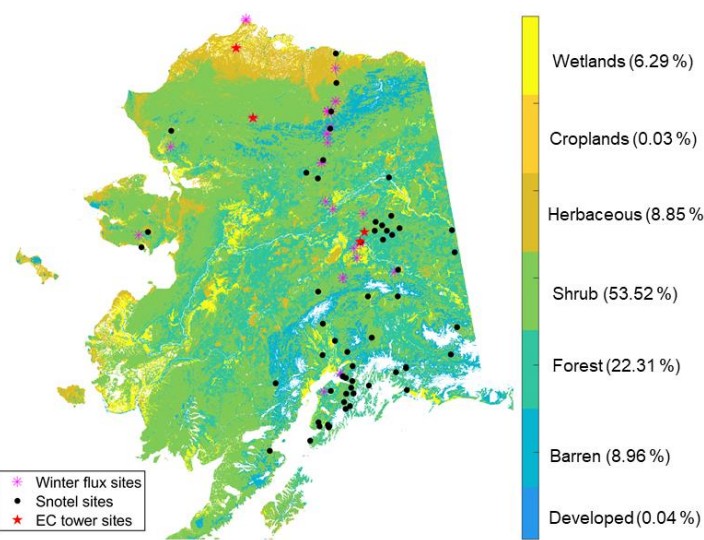

**Fig. 1** The Alaskan land cover map and the location of in-situ sites used for model validation. The land cover types are aggregated
from the 30-m NLCD map (Jin et al., 2013), while the following land cover classes were used in the model simulations: developed
and barren land, forest, scrub/shrub, grassland/herbaceous, croplands, and wetlands. The percentage of each land cover type was
provided alongside the labels of the colorbar.



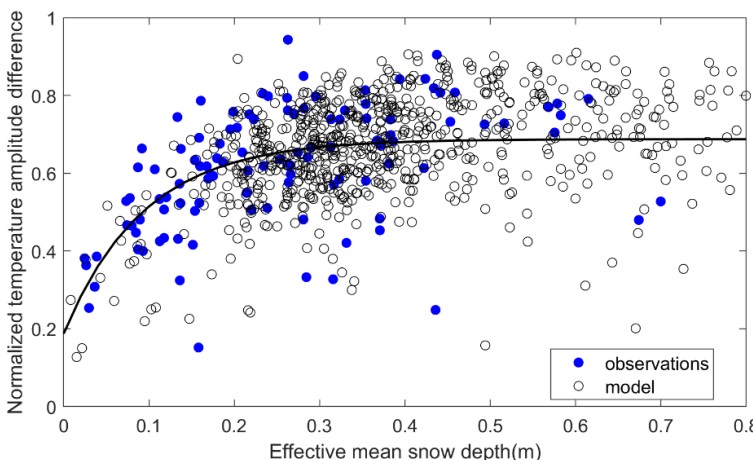

**Fig. 2** Comparison of the snow insulation curve derived from in-situ measurements and the model simulations at Alaskan Snotel sites. The dark line is drawn using the parameters presented in Slate et al. (2017):    $A_{norm} = 0.1875 + 0.5 \times (1 - e^{-(S_{depth,eff}/0.0941)})$. Observations have fewer data points due to data gap in the observed snow depth and soil temperatures at the Snotel sites.





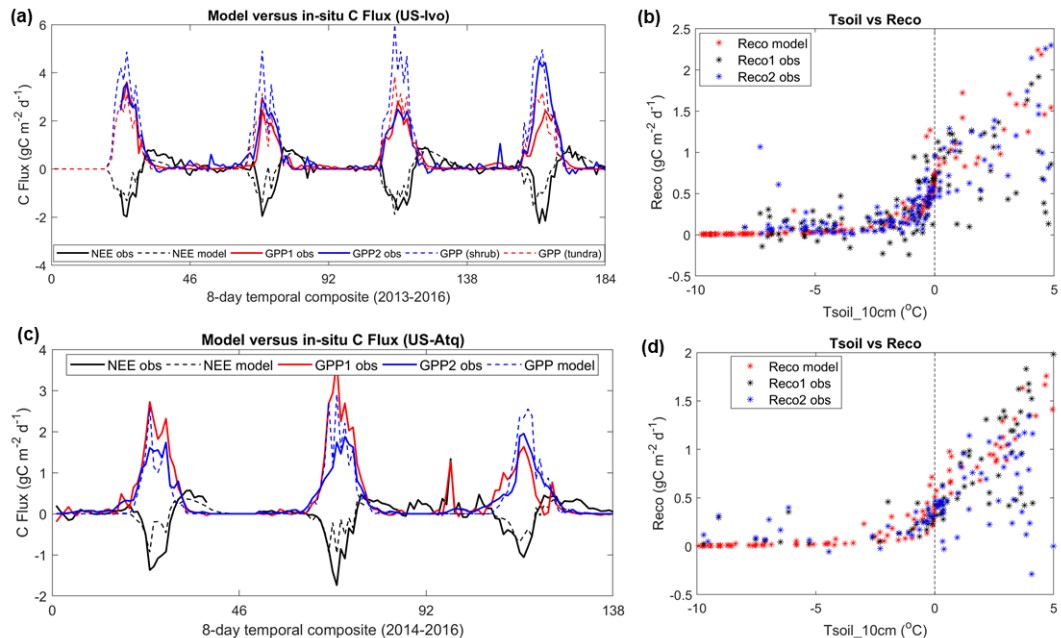


**Fig. 3** Model simulated carbon fluxes and temperature sensitivity of ecosystem respiration at two tundra sites (US-Ivo: a-b; US-Atq: c-d). "GPP1 obs" and "GPP2 obs" represent GPP estimates derived using tower-based NEE measurements and different partitioning methods provided by the tower PI, similar as "Reco1 obs" and "Reco2 obs". At US-Ivo site, two GPP simulations were conducted using different maximum LUE parameters of two vegetation types (shrub and tundra), indicated as "GPP
(shrub)", and "GPP (tundra)" in panel (a).



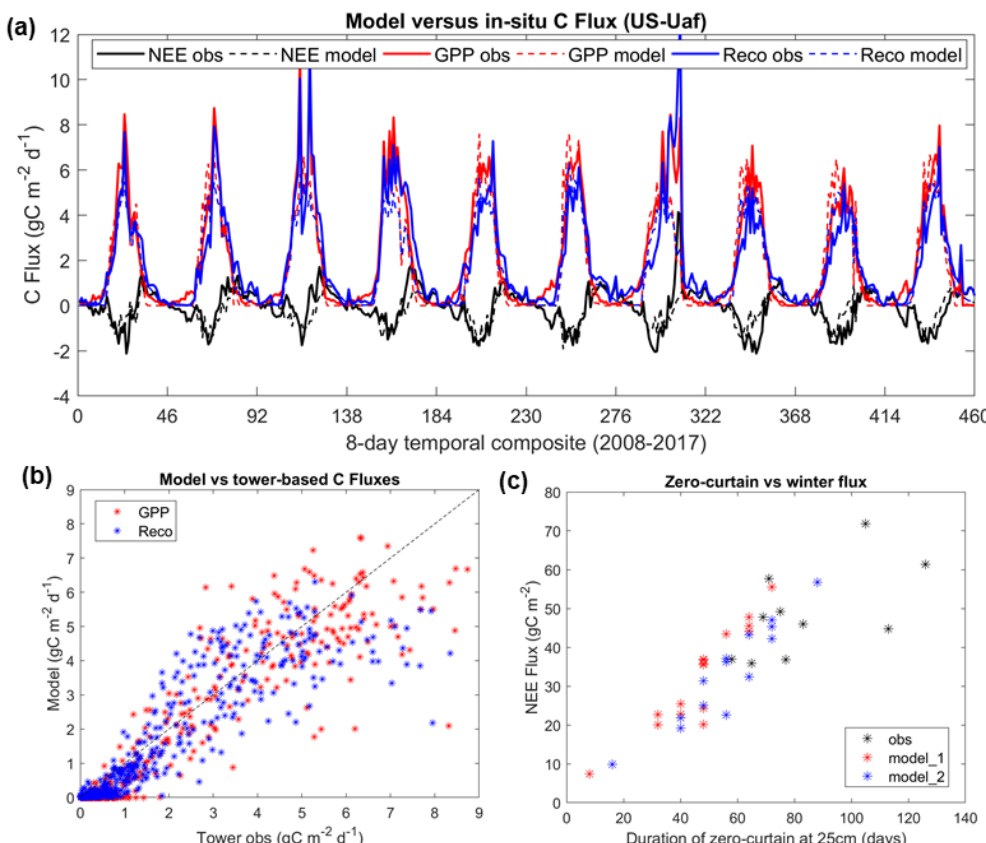

**Fig. 4** Comparisons of model simulated carbon fluxes with tower-based estimates (a&b), and the relations of total NEE fluxes to the zero-curtain duration at 25 cm soil depth (c). There was a significant correlation (R = 0.6, p < 0.1) between the zero-curtain period derived from in-situ soil moisture data and the total NEE fluxes during this period. "model_1" and "model_2" indicate model simulations using different soil saturation levels, with "model_2" using a slightly higher (120%) saturation level than "model_1".



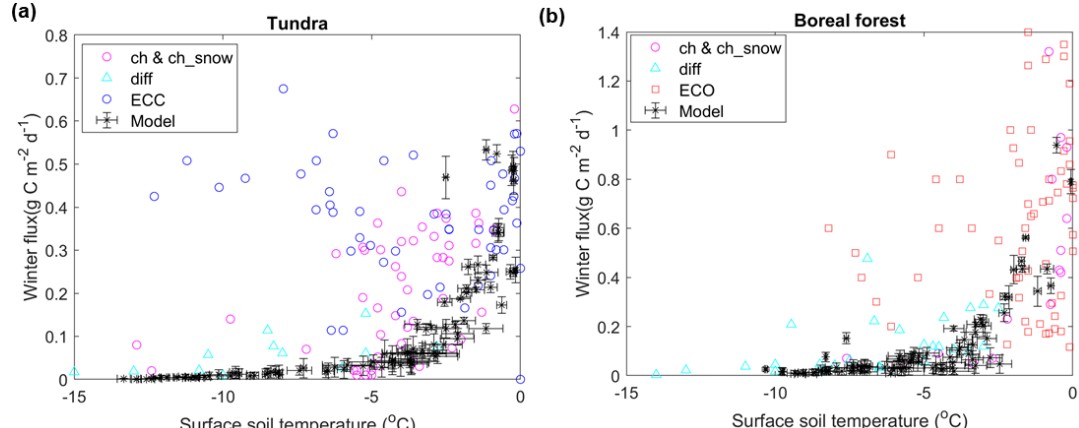

**Fig. 5** Effects of soil temperature on $CO_2$ fluxes during the cold season over Alaskan tundra (a) and boreal forest (b) sites indicated by model simulations (this study) and in-situ observations from a winter flux synthesis dataset (Natali et al., 2019b). "ch&ch_snow", "diff", "ECC" and "ECO" represent measurements made using chambers and chambers placed atop the snowpack, diffusion, EC-closed path, and EC-open path methods respectively. The error bars indicate the standard deviations of model simulations using different values (0.35 ~ 0.9) for the dimensionless parameter characterizing the unfrozen water curve for most soil types (Schaefer and Jafarov, 2016).






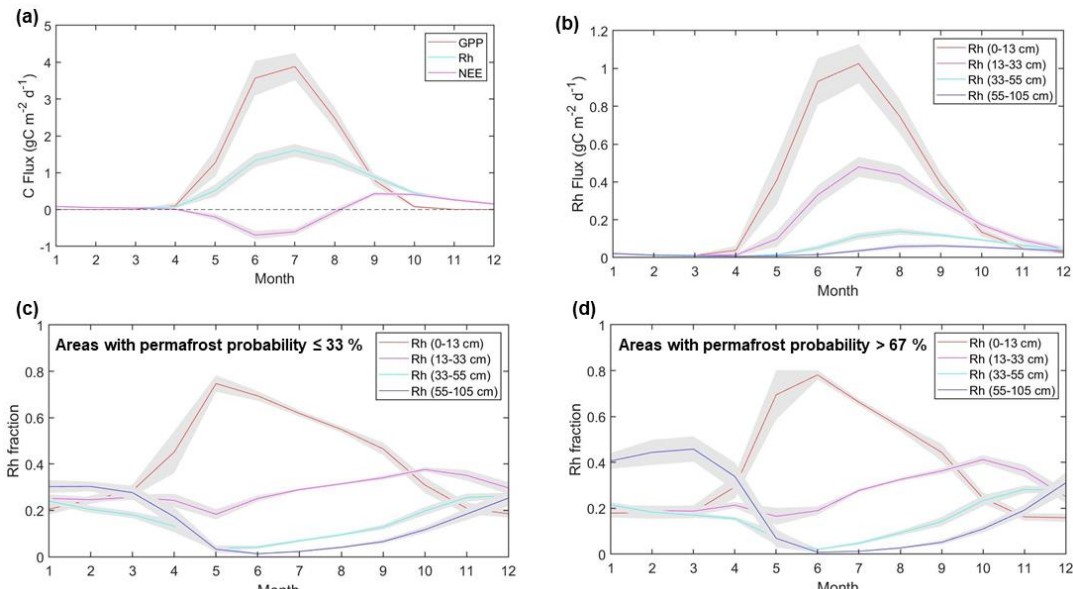


**Fig. 6** Regional mean of model simulated carbon fluxes (a), Rh fluxes from different soil depths (b) averaged across Alaska, and Rh contribution from different soil depths to total Rh in areas with low (c) and high (d) permafrost probability. Gray shading areas denote the standard deviation of monthly mean fluxes from 2001 to 2017.


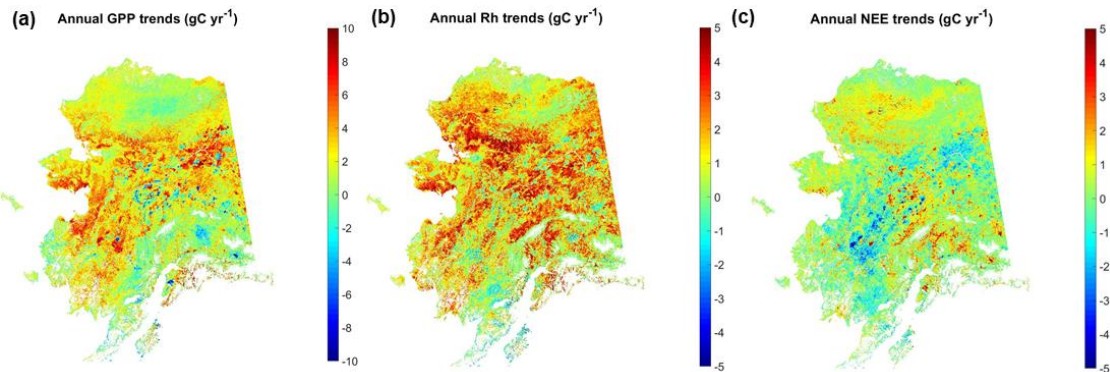

**Fig. 7** Temporal trends of model estimated annual carbon fluxes from 2001 to 2017. For NEE, positive trends indicate decreasing net carbon uptake activity, while negative trends indicate enhanced net carbon uptake.



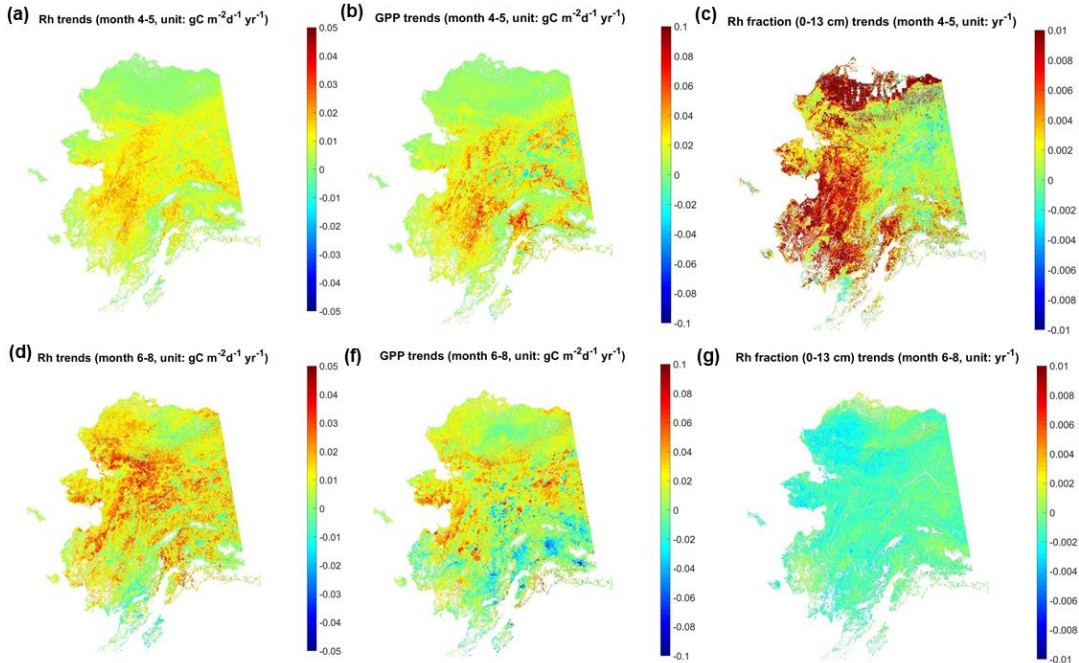

**Fig. 8** Temporal trends of model estimated total Rh, GPP and surface soil contribution to Rh (Rh fraction) during the early and peak growing season from 2001 to 2017. In panel (c), large areas in the Alaska North Slope were masked out (in white) due to negligible total Rh fluxes in April.



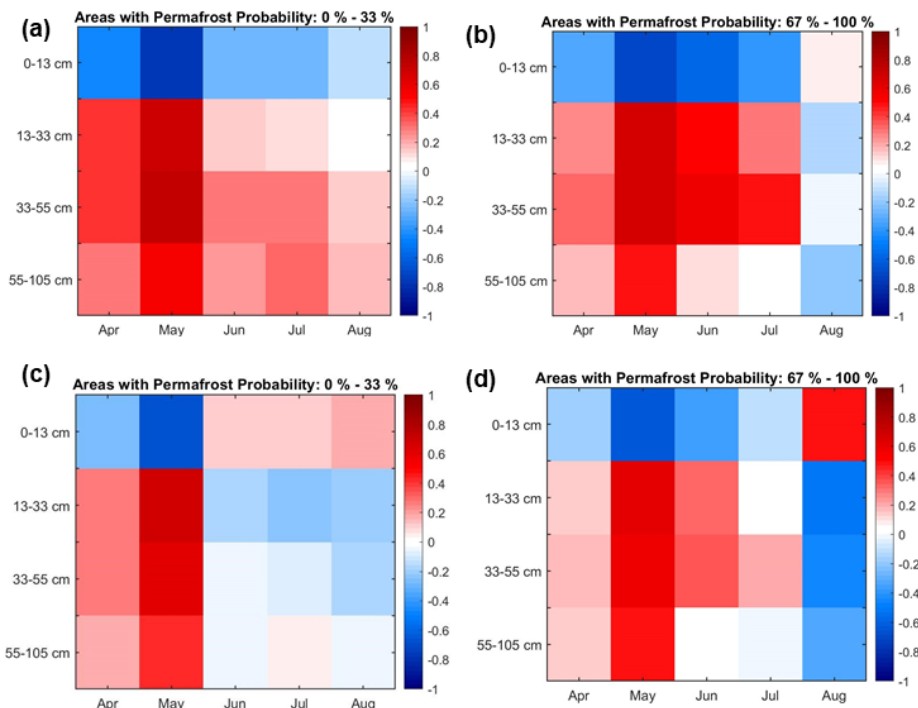


**Fig. 9** Correlation analysis between model estimated Rh fraction from different soil depths during summer season (June-August) and monthly MODIS LST for different permafrost regions: (a-b): Pearson correlation coefficient; (c-d) partial correlation between LST and Rh fraction, using growing-season (April-August) GPP as the controlling variable.




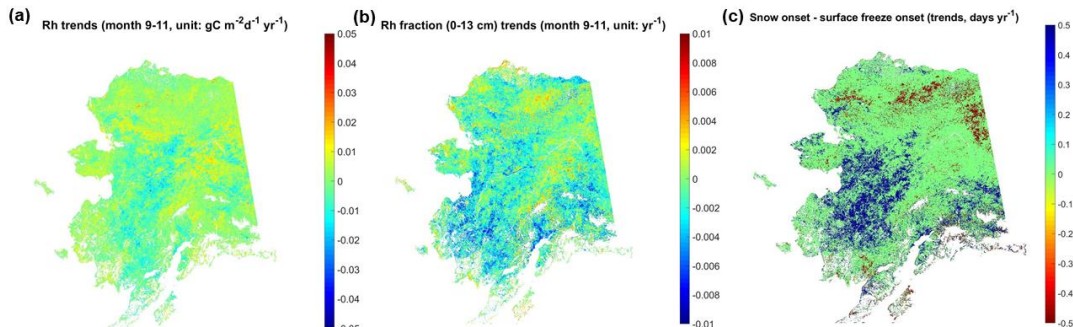

**Fig. 10** Regional trends of total Rh (a) and its surface soil contribution (b) during the early cold season (September-November) versus regional trends of the number of snow-free days after land surface freezes (c), which was defined as the difference between snow onset and surface freeze onset.

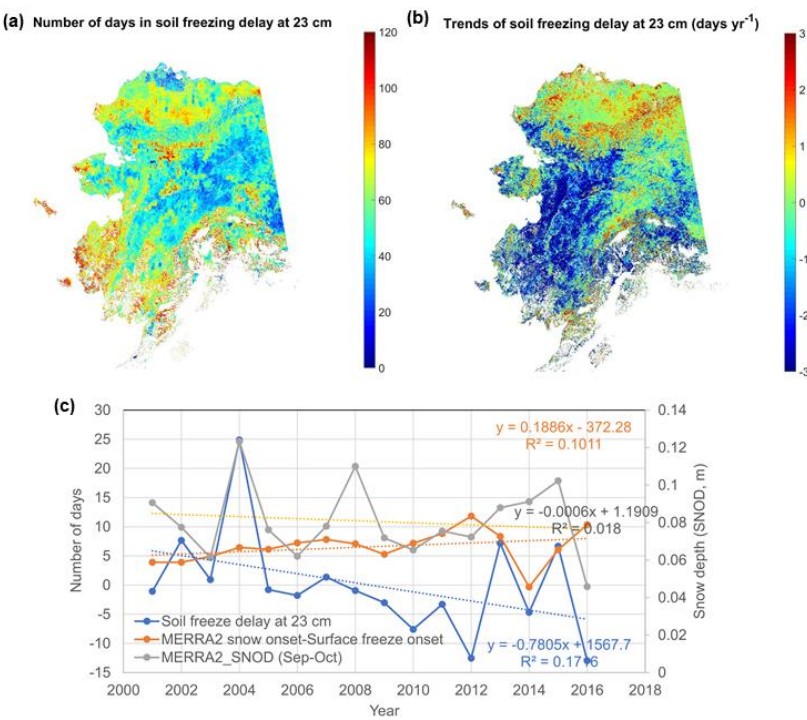


**Fig. 11** Sensitivity of model simulated soil freezing process to snow cover changes across Alaska: the mean (a) and trends (b) of soil freezing delay at 23 cm soil depth relative to surface freeze onset; c) the annual time series of model simulated soil freezing delay, the number of snow-free days after land surface freezes, and MERRA-2 snow depth (SNOD) from September to October averaged across Alaska.

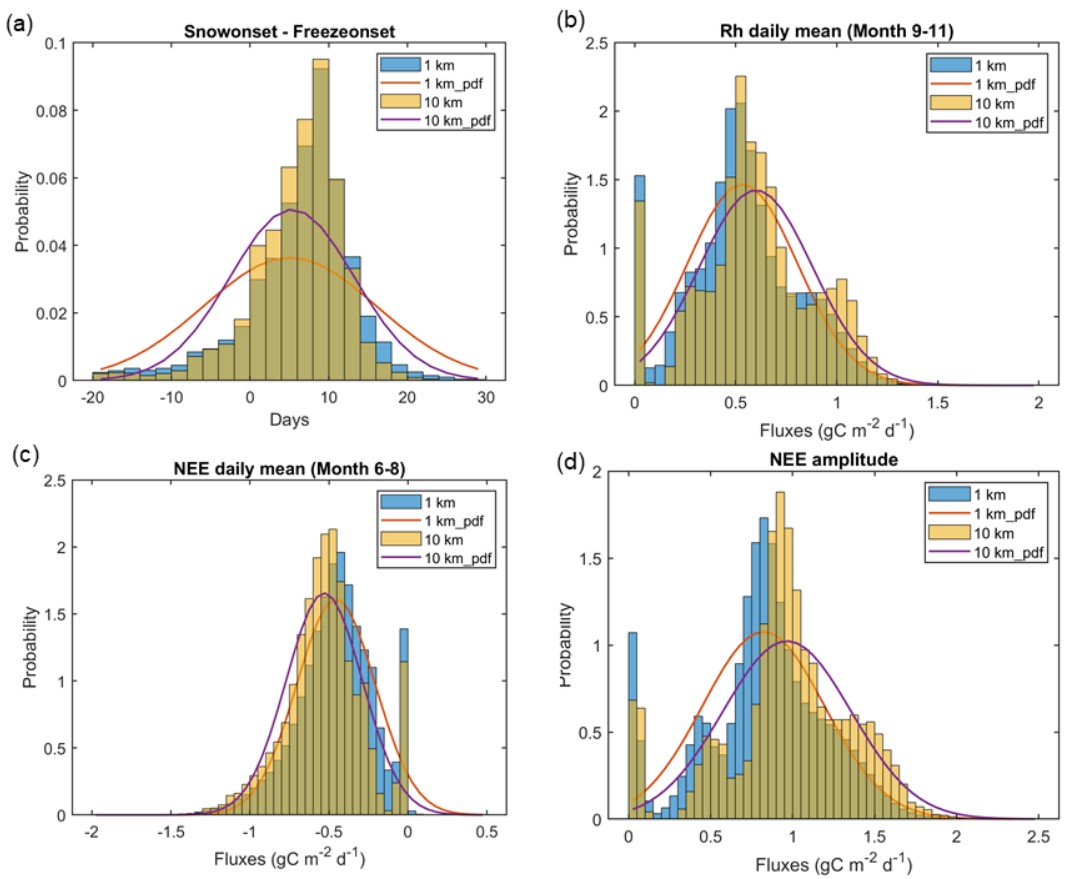

**Fig. 12** Comparisons of statistical distribution of model inputs and simulations at 1-km and 10-km resolution across Alaska: (a) the number of snow-free days after land surface freeze onset, derived from the model input LST and snow depth datasets; (b) model simulated daily mean Rh flux averaged from September to November; (c) model simulated daily mean NEE flux averaged from June to August; (d) model simulated NEE amplitude, which was defined as the difference of the daily mean NEE flux between two periods (September-November vs June-August). The lines show the fitted probability distribution function (pdf) using a normal distribution.