# Peer review of "Investigating the sensitivity of soil heterotrophic respiration to recent snow cover changes in Alaska using a satellite-based permafrost carbon model"

_Biogeosciences, 2020_

## Referee Comment (RC1) · Anonymous Referee #1 · 26 Jul 2020

General comments

This study describes a novel permafrost carbon model, with two distinguishing features: it's largely driven by remote sensing data, and operates at an intermediate spatial scale. The authors describe the model and its parameterizations clearly, test it against a number of eddy covariance and distributed datasets across Alaska, and then use it to predict regional fluxes. This is an important and interesting subject with wide possible implications. The ms is generally well written and interesting; figures are clear; and the introduction does a very nice job of setting up the overall study.

There are some problems. The results are wordy and fairly long; one problem is that

there's a certain amount of discussion material mixed in. I suggest looking for opportunities to condense and cleanly separate different sections' material. I also was quite confused how you're comparing model output Rh with the Natali dataset, which is soil surface (Ra+Rh); there's a general carelessness with terminology in this area, confusing the reader about whether soil surface $CO_2$ flux (soil respiration) or its heterotrophic component is being referred to. Finally, it's not acceptable, in my view, not to make the model code available at the review stage. For all these see below.

In summary, this is overall a strong, interesting, and well-done study. It would benefit from moderate revisions for clarity and concision in many places, and transparency and reproducibility absolutely need to be improved.

Specific comments

1. Line 30: "soil respiration" or heterotrophic respiration? I assume we're still talking about the latter, but clarify. Similarly line 31 mentions "total soil carbon emissions" – is this the Ra+Rh flux at the surface?

2. L. 71: define soil respiration precisely here

3. L. 107: how are these depths chosen?

4. L. 125: "linear"?

5. L. 129: interesting assumption. What's the rationale? Does litterfall = 100% of NPP in other systems, or at regional research sites?

6. L. 265: "therefore. . ." this logic is unclear. How the 2001-2016 period related to first part of sentence?

7. L. 301-303: this sentence seems out of place

8. L. 304: perhaps start new paragraph here

9. L. 306-312: seems like discussion, not results

10. L. 345: I'm confused how you're comparing model output Rh with the Natali dataset, which is soil surface (Ra+Rh)

11. L. 522-531: this seems unnecessary and duplicative of conclusions below

12. L. 560: perhaps start new paragraph for readability

13. L. 606-608: it's really inexcusable, in my view, to promise to upload data and code in the future while not making it available at the review stage

---

## Referee Comment (RC2) · Anonymous Referee #2 · 1 Aug 2020

In this study, Yi et al used a satellite-based permafrost carbon model to analyze the response of soil respiration to changes in snow coverage and temperature in the Alaska ecosystems. They concluded that for the time period from 2001 to 2017, soil respiration has overall increased with the warming. While I can sense the study was well-attempted and carefully written, I feel some additional analyses may further improve the quantitative strength of some of the currently too colloquial conclusions. For instance, a time series plot showing how the carbon fluxes over Alaska have changed through the whole time period will give readers a more direct visual impression. In addition, in the trend analysis presented in Fig. 7, it is unclear how such trend change should be put into the context of changes in snow cover and warming. Perhaps an attribution

analysis of these carbon flux trends to changes in snow cover, temperature, ALT, etc. will be helpful? Finally, maybe the authors could think of beginning the paper with a diagram (or flow chart) of how soil respiration is related to the variables they are investigating in this study? Such a diagram will put the attribution analysis (if the authors decide to do it) or the analysis in the result section into a better mental perspective. Some more specific points are given below.

Model description: the hydrological module is not well described. It took me quite a while to figure out the soil moisture is not simulated but rather is model input (am I right?). Moreover, in order to understand the model, I also read a number of other papers about the model, but was never clear how the whole model was assembled. So, if I may request, can the authors present a model description as supplemental material? Or at least give a list of what major variables are simulated, and what are prescribed as input.

Fig 2, it is not easy to compare model with observations, even though I can see the model ball-park agrees with the response curve derived in Slate et al. (2017). The authors may consider interpolate the model results to the observations and present a scatter-plot as an addition to help analyzing the model performance.

Fig 3, it will be helpful to present a scatter-plot of modeled vs measured NEE.

Fig 6. Panel c and d are hard to compare, maybe the authors can consider contrasting two depths each panel in two panels, so readers can compare the time series more straightforwardly.

Fig. 7, like in my major comments, if a quantitative attribution analysis can be done here, it will be very helpful.

Other minor comments: L 204 "soil moisture" is unclear, maybe "liquid water" should be used.

---

## Author Comment (AC1) · 23 Aug 2020

**Response to referee's comments on "Investigating the sensitivity of soil respiration to recent snow cover changes in Alaska using a satellite-based permafrost carbon model"**

**Dear Editor,**

We appreciate the constructive comments from the two reviewers, and have carefully revised the paper based on those comments. Specifically, we added a flowchart and a brief model description to make the paper easier to follow; we performed attribution analysis to investigate main climate controls on annual carbon fluxes. We also paid particular attention on the definition of soil respiration and removed "redundant" discussion throughout the results and discussion section. Our responses to the comments are provided in the following text, and the revised manuscript is enclosed as a supplement with changes highlighted.

Thank you very much for considering our manuscript.

**Yonghong Yi**, on behalf of all authors

**Review 1#:**

1) General comments: "There are some problems. The results are wordy and fairly long; one problem is that there's a certain amount of discussion material mixed in. I suggest looking for opportunities to condense and cleanly separate different sections' material. I also was quite confused how you're comparing model output Rh with the Natali dataset, which is soil surface (Ra+Rh); there's a general carelessness with terminology in this area, confusing the reader about whether soil surface CO2 flux (soil respiration) or its heterotrophic component is being referred to. Finally, it's not acceptable, in my view, not to make the model code available at the review stage. For all these see below.
In summary, this is overall a strong, interesting, and well-done study. It would benefit from moderate revisions for clarity and concision in many places, and transparency and reproducibility absolutely need to be improved."

**Response:**

Thank you for the comments. We have carefully gone through the Results session (Section 3), and removed redundant discussion materials in Section 3 or moved them into Section 4, in order to be more concise, including section 3.1.1, 3.1.2, 3.2.2. Please refer to the manuscript for more details. We also redefined the "soil respiration" in the paper, and made it clear how we compared the model simulations with Natali's dataset. Please see the details in our response below. Finally, the code now is made public on GitHub: https://github.com/yiyh05/STM-C.

2) Line 30: "soil respiration" or heterotrophic respiration? I assume we're still talking about the latter, but clarify. Similarly line 31 mentions "total soil carbon emissions" – is this the Ra+Rh flux at the surface?

**Response:**

Both here we are talking about "soil heterotrophic respiration". We revised the abstract and the title to be more specific.

3) L. 71: define soil respiration precisely here

**Response:**

We revised the text to clarify this:

Line 72-76: "Soil respiration is mainly the product of respiration by roots (autotrophic) and soil decomposers (heterotrophic); however, it is generally difficult to partition it into its heterotrophic and autotrophic source (Phillips et al., 2017). In this study, we focus on the heterotrophic component of soil respiration, and assume it is the dominant component of total soil respiration in northern ecosystems during the cold season due to root dormancy (Tucker et al., 2014; Hicks Pries et al., 2015)."

4) L. 107: how are these depths chosen?

**Response:**

The model soil depth definition follows the previous model setup (Rawlins et al., 2013; Yi et al., 2015), with fine resolution at the soil surface and increasing thickness along the soil depth. The depth setup can change as long as having a finer vertical resolution at the surface, in order to ensure the stability of solving the partial differential equations using finite-difference numerical methods, including both the 1-D soil heat transfer equation (Eq. S1) and soil carbon transport equation (Eq. 3). Please note that we now moved this sentence to the model description in the supplementary material (S1).

5) L. 125: "linear"?

**Response:**

Yes, it was corrected.

6) L. 129: interesting assumption. What's the rationale? Does litterfall = 100% of NPP in other systems, or at regional research sites?

**Response:**

Disturbances can significantly alter the balance between annual NPP and litterfall, which may be a large uncertainty source to our simulated NEE and respiration fluxes. In the northern ecosystem, fire disturbance is likely the largest contributing factor to this uncertainty. In other ecosystems, land use change and other disturbance events, such as harvest, insect etc, can be also important depending on the region. However, modelling disturbance effects on carbon balance at regional scale is generally a challenge. In addition, please note that litterfall in this study also includes the carbon turnover from the woody components with a low turnover rate. The woody fraction from different land cover types can vary from 10% to 40% (Table S1), which was generalized from parameters used in BIOME-BGC model and data collected in White et al. (2000).

7) L. 265: "therefore: : :" this logic is unclear. How the 2001-2016 period related to first part of sentence?

**Response:**

We clarified this in the main text:

Line 279-282: "Unfrozen conditions at deep active layer may persist well into the cold season and even into January, and the soil freeze onset at deep depths in the current year may occur in the next calendar year. Since the model was only run from 2001 to 2017, the soil freeze onset and delay in year 2017 were not calculated."

8) L. 301-303: this sentence seems out of place

**Response:**

The differences in the model simulated GPP at the US-Ivo site using different biome types ("Shrub" vs "Tundra", Fig. 4) were mainly due to different maximum light use efficiency ($\varepsilon_{max}$) values specified for the two plant function types (Table S1). This parameter ($\varepsilon_{max}$) can show large variability across and within plant function types (Madani et al., 2014). However, we agree that this belongs to discussion, and we removed this sentence to be more concise.

9) L. 304: perhaps start new paragraph here

**Response:**

We broke the paragraph into two as suggested by the reviewer, due to a very long paragraph. Please refer to Line 334-344 for details.

10) L. 306-312: seems like discussion, not results

**Response:**

This was indeed discussion on what caused the mismatch between model and simulated carbon fluxes. However, the uncertainties in the MODIS LST, and the eddy covariance tower NEE partition method (mainly occur during the growing season) discussed here were not the focus of the discussion section. Therefore, we think it is better to place them here. However, we revised these sentences to be more concise:

Line 336-339: "The largest GPP reductions during the peak season were generally caused by very low nighttime LST, which may have large uncertainties in cloudy sky conditions. In addition, there is also large uncertainty imposed from the NEE partitioning method, where different methods result in large differences (up to more than 1 g C m$^{-2}$ d$^{-1}$) in the tower-based GPP and Reco estimates."

11) L. 345: I'm confused how you're comparing model output Rh with the Natali dataset, which is soil surface (Ra+Rh)

**Response:**

We acknowledged the differences between Rh and the winter soil $CO_2$ flux from the Natali dataset, which includes both soil autotrophic and heterotrophic respiration. It would be more consistent if we directly compared the model simulated and measured Rh fluxes. However, there are very limited studies that provided reliable general partitioning methods between the autotrophic and heterotrophic respiration for the northern ecosystems.

Yet, our comparison may still make sense based on the following considerations. First, the comparison was made only during the cold season defined from October from April. During this period, model simulated GPP is generally very low (especially for tundra); therefore, the model simulated total respiration is close to Rh. Secondly, a few studies have also showed that Rh is the

dominant component of total soil respiration in alpine and northern ecosystems during the cold season (Du et al., 2013; Tucker et al., 2014; Hicks Pries et al., 2015). In addition, our comparison focuses on the temperature sensitivity of carbon fluxes during the cold season between the model and the in-situ data, rather than directly compare the two fluxes.

We added a sentence in the methods to clarify this:

Line 240-242: "In this study, we compared the model simulated soil heterotrophic respiration directly with the measured soil $CO_2$ flux, since model simulated autotrophic respiration (as a portion of GPP) is very low throughout the cold season, especially for the tundra."

**12) L. 522-531: this seems unnecessary and duplicative of conclusions below**

**Response:**

We removed this paragraph for brevity as suggested.

**13) L. 560: perhaps start new paragraph for readability**

**Response:**

We separated the paragraph into two as suggested by the reviewer. Please refer to Line 587 for details.

**14) L. 606-608: it's really inexcusable, in my view, to promise to upload data and code in the future while not making it available at the review stage**

**Response:**

The code now is made public on GitHub: https://github.com/yiyh05/STM-C. The data produced by this study will be submitted to ORNL DAAC as part of the NASA ABoVE archive; however, please note that it generally takes a long time for ORNL to review, accept and publish the dataset.

**References:**

Du, E., Zhou, Z., Li, P., Jiang, L., Hu, X. and Fang, J.: Winter soil respiration during soil-freezing process in a boreal forest in Northeast China, Journal of Plant Ecology, 6(5), 349–357, doi:10.1093/jpe/rtt012, 2013.

Madani, N., Kimball, J. S., Affleck, D. L. R., Kattge, J., Graham, J., van Bodegom, P. M., Reich, P. B. and Running, S. W.: Improving ecosystem productivity modeling through spatially explicit estimation of optimal light use efficiency: Ecosystem optimal light use efficiency, J. Geophys. Res. Biogeosci., 119(9), 1755–1769, doi:10.1002/2014JG002709, 2014.

White, M. A., Thornton, P. E., Running, S. W., and Nemani, R. R.: Parameterization and Sensitivity Analysis of the BIOME–BGC Terrestrial Ecosystem Model: Net Primary Production Controls, Earth Interact., 4, 1–85, 2000.

Rawlins, M. A., Nicolsky, D. J., McDonald, K. C. and Romanovsky, V. E.: Simulating soil freeze/thaw dynamics with an improved pan-Arctic water balance model: SOIL FREEZE/THAW MODELING, J. Adv. Model. Earth Syst., 5(4), 659–675, doi:10.1002/jame.20045, 2013.

Yi, Y., Kimball, J. S., Rawlins, M. A., Moghaddam, M. and Euskirchen, E. S.: The role of snow cover affecting boreal-arctic soil freeze–thaw and carbon dynamics, Biogeosciences, 12(19), 5811–5829, doi:10.5194/bg-12-5811-2015, 2015.

Phillips, C. L., Bond-Lamberty, B., Desai, A. R., Lavoie, M., Risk, D., Tang, J., Todd-Brown, K. and Vargas, R.: The value of soil respiration measurements for interpreting and modeling terrestrial carbon cycling, Plant Soil, 413(1–2), 1–25, doi:10.1007/s11104-016-3084-x, 2017.

Hicks Pries, C. E., van Logtestijn, R. S. P., Schuur, E. A. G., Natali, S. M., Cornelissen, J. H. C., Aerts, R. and Dorrepaal, E.: Decadal warming causes a consistent and persistent shift from heterotrophic to autotrophic respiration in contrasting permafrost ecosystems, Glob Change Biol, 21(12), 4508–4519, doi:10.1111/gcb.13032, 2015.

Tucker, C. L., Young, J. M., Williams, D. G. and Ogle, K.: Process-based isotope partitioning of winter soil respiration in a subalpine ecosystem reveals importance of rhizospheric respiration, Biogeochemistry, 121(2), 389–408, doi:10.1007/s10533-014-0008-9, 2014.

**Review 2#:**

1) General comments: "In this study, Yi et al used a satellite-based permafrost carbon model to analyze the response of soil respiration to changes in snow coverage and temperature in the Alaska ecosystems. They concluded that for the time period from 2001 to 2017, soil respiration has overall increased with the warming. While I can sense the study was well attempted and carefully written, I feel some additional analyses may further improve the quantitative strength of some of the currently too colloquial conclusions. For instance, a time series plot showing how the carbon fluxes over Alaska have changed through the whole time period will give readers a more direct visual impression. In addition, in the trend analysis presented in Fig. 7, it is unclear how such trend change should be put into the context of changes in snow cover and warming. Perhaps an attribution analysis of these carbon flux trends to changes in snow cover, temperature, ALT, etc. will be helpful? Finally, maybe the authors could think of beginning the paper with a diagram (or flow chart) of how soil respiration is related to the variables they are investigating in this study? Such a diagram will put the attribution analysis (if the authors decide to do it) or the analysis in the result section into a better mental perspective."

**Response**:

Thank you for the suggestion. In response to your comments, we have: 1) added a time series plot (Fig. S10), and performed additional attribution analysis (Fig. 9) to support our conclusion; and 2) added a diagram of the modelling framework (Fig. 1). We also added description on the permafrost soil model that we used and a few figures in the supplementary materials to address the reviewer's concerns. Please see our response below for more details.

2) Model description: the hydrological module is not well described. It took me quite a while to figure out the soil moisture is not simulated but rather is model input (am I right?). Moreover, in order to understand the model, I also read a number of other papers about the model, but was never clear how the whole model was assembled. So, if I may request, can the authors present a model description as supplemental material? Or at least give a list of what major variables are simulated, and what are prescribed as input.

**Response**:

Yes, the permafrost soil model (RS-PM) does not simulate the soil water movement directly; rather it uses the total soil water content from SMAP L4SM product as the inputs, and then simulates the changes in the unfrozen liquid water fraction due to soil freeze/thaw activity. We have added a short paragraph in the beginning of Section 2.1 to more clearly illustrate the modeling process and the link between the permafrost soil model and the carbon model:

Line 103-112: "The Remote Sensing driven Permafrost Model (RS-PM) developed in Yi et al (2018; 2019), was coupled with a terrestrial carbon flux (TCF) model (Yi et al., 2015) to investigate the climate sensitivity of carbon fluxes across Alaska (Fig. 1), with a particular focus on the shoulder season. The soil decomposition model in the original TCF model was revised in this study to account for vertical soil carbon transport in order to better simulate the depth-dependent soil carbon distribution and respiration fluxes. The RS-PM model simulates the soil temperature and changes in soil liquid water content due to soil freeze/thaw along the soil profile,

using remote sensing datasets including land surface temperature (LST), snow cover information and total soil moisture content. The RS-PM outputs were then used as inputs to the TCF model, as constraints on both the vegetation productivity and soil respiration. A brief description of the modeling framework was described here, with a focus on the revised soil decomposition model, while a detailed description on the RS-PM model was provided in the supplementary material."

The flow diagram was presented as the new Figure 1. We also provided description on the permafrost soil model in the supplementary materials. Please refer to the manuscript for more details.

[Figure]

**Fig. 1** Flow diagram describing the modelling procedure and main input datasets used in this study. The terrestrial carbon flux model has two components, including the light use efficiency algorithm for vegetation productivity estimates and a soil decomposition model for soil heterotrophic respiration estimates. The main equations used for each modelling component was also included in the modelling box.

3) Fig 2, it is not easy to compare model with observations, even though I can see the model ball-park agrees with the response curve derived in Slate et al. (2017). The authors may consider interpolate the model results to the observations and present a scatter-plot as an addition to help analyzing the model performance.

**Response**:

There is a generally large discrepancy between downscaled MERRA2 (1-km) and in-situ effective snow depth data at the Snotel sites (Fig. S1a), so we chose not to directly compare the

model simulated and in-situ soil temperature data at the Snotel sites. But we do see an overall consistency between model simulated soil temperature and in-situ data at 20 cm depth as shown in Fig. S1b. Soil temperature data at 5 and 50 cm show similar performance. We chose not to include this figure in the main text but in the supplementary material to make the paper more concise.

[Figure]

**Fig. S1** Comparison between (a) effective snow depth derived from in-situ observations and downscaled MERRA2 data, and (b) observed and model simulated monthly soil temperature at 20 cm at the Snotel sites. Note that the sites compared for snow depth and soil temperature are not the same due to inconsistency between the snow depth and soil temperature measurements at the Snotel sites. There are generally more snow depth measurements than soil temperature measurements at the Snotel sites.

4) Fig 3, it will be helpful to present a scatter-plot of modeled vs measured NEE.

**Response**:

The scatter-plots between modeled and measured NEE fluxes are now added as panel (d) in Fig. 4 (the original Fig. 3). The temperature sensitivity of ecosystem respiration at US-Atq in the original panel (d) was now presented as Fig. S2.

5) Fig 6. Panel c and d are hard to compare, maybe the authors can consider contrasting two depths each panel in two panels, so readers can compare the time series more straightforwardly.

**Response**:

According to the reviewer's suggestion, we now combined panel (c) and (d) in Fig. 7 (originally as Fig. 6) as a single panel, and compared the depth-dependent Rh fraction for the two permafrost zones. We combined the two intermediate soil depths (13-33 cm, 33-55 cm) as a single depth (13-55 cm), to be more concise. Please refer to the new Figure 7 for more details.

6) Fig. 7, like in my major comments, if a quantitative attribution analysis can be done here, it will be very helpful.

**Response**:

We added two figures to support quantitative analysis of our results as requested by the reviewer: 1) Fig. S10 shows the time series plot of the annual carbon fluxes; 2) Fig. 9 shows the relative importance of selected climate variables to the annual carbon fluxes. The original Fig. 9 that provides results on the correlation analysis between Rh fraction and seasonal LST was now moved to the supplementary material (Fig. S13) to make the paper more concise.

The attribution analysis was conducted using the gradient boosting regression method, and was described in Section 2.4 (Line 289-302):

"Finally, we used the gradient boosting regression (GBR) method to quantify the contribution of climate variables to the annual carbon fluxes. The GBR method consists of a sequence of models, and each consecutive model is developed based on the errors of previously added models (Friedman, 2000). The above model simulated annual carbon fluxes from 2002 to 2017 were used to train and evaluate the GBR models. We chose the following nine contributing factors or predictors to annual carbon fluxes during the model fitting, including summer (June-August) NDVI, annual freezing and thawing index, mean annual downward solar radiation, rootzone soil moisture during the thaw season, snow offset and onset, mean snow depth averaged from January to March (representing annual maximum snow depth), and snow depth during the early snow season (from October to November). The GBR method was implemented using the sklearn package of Python 2.7. The following method was used to determine the relative importance of each predictor to the GBR model's predictive performance. We first run the GBR model using all nine predictors, and the model results were referred as baseline simulation ($GBR_{baseline}$). We then ran the fitted model with one randomized variable but with other variables remained intact, and the results were referred as $GBR_{one\_variable\_randomized}$. The variable importance was then computed and normalized based on the Person's correlation coefficient between the two runs using the following equations (Karjalainen et al., 2019; Zheng et al., 2020):

$$I_x = 1 - \mathrm{corr}(\mathrm{GBR_{baseline}} - \mathrm{GBR_{one\_variable\_randomized}})$$
$$RI_x = \frac{I_x}{\sum_{x=1,9} I_x} \tag{6}$$
"

The attribution analysis results were added in section 3.2.1 (Line 413-430):

"…At the regional scale, the time series of annual carbon fluxes also showed non-significant ($p>0.1$) positive trends, with values of 2.58, 1.86, 0.38 Tg C yr$^{-1}$ for GPP, Rh and NEE fluxes respectively (Fig. S10).

The attribution analysis results using the GBR method also indicate that summer NDVI and annual thawing index are the two most important variables affecting the annual carbon fluxes, which was generally consistent across different vegetation types (Fig. 9). For annual GPP flux, NDVI was the most important variable followed by annual thawing index and downward solar radiation, while for annual Rh fluxes, annual thawing index was the most important variable, followed by NDVI, with other variables playing a very minor role. Despite the importance of annual thawing index controlling annual GPP and Rh fluxes, the snow offset showed little importance to both fluxes. This was likely due to the low temporal resolution of the MODIS snow cover data used for the snow offset calculation, which was calculated as the center date of the 8-day composite period with snow disappearance. The low temporal resolution of snow

offset (i.e. discrete variables) and a strong correlation (R>0.7, p<0.1) between annual thawing index and snow offset may limit its use in the regression model. As for annual NEE flux, thawing index, NDVI, downward solar radiation, and annual freezing index are among the most important factors. However, the effects of different variables on annual NEE flux varied throughout the period due to their compensating effects on GPP and NEE, and NEE being a small residual flux; therefore, none of the variables played a dominant role throughout the entire period. The GBR model also showed a relatively poor performance in prediction of annual NEE fluxes (R ≥ 0.7) comparing with the other two fluxes (R > 0.9)."

[Figure]

**Fig. 9** Mean relative importance values of climate variables in controlling annual carbon fluxes in Alaska (a: GPP; b: Rh; c: NEE). The importance values were averaged for four major vegetation types (Forest, Shrub, Herbaceous, and Wetlands, Fig. 2), and the error bar represents their standard deviation across different vegetation types. The nine variables are: summer (June-August) NDVI, annual thawing and freezing index, snow offset and onset, mean snow depth averaged from January to March (representing annual maximum snow depth, SNODmax), and snow depth averaged during the early snow season (from October to November, SNOD_fall), mean annual downward solar radiation, and rootzone soil moisture during the thaw season. The annual thawing and freezing index are the sum of MODIS LST above 0 °C and below 0 °C throughout the year respectively.

7) Other minor comments: L 204 "soil moisture" is unclear, maybe "liquid water" should be used.

**Response**:

We now use "liquid water content" instead of "soil moisture".

---

## Author Comment (AC3) · 23 Aug 2020

==highlight==

**I. Model description**

**(1) The remote sensing driven permafrost model (RS-PM)**

The remote sensing based permafrost model, as described in Yi et al. (2018; 2019), uses a numerical approach to simulate soil temperature and soil freeze/thaw (F/T) process (and changes in soil liquid water content) along the 60 m soil profile using 23 soil layers, with finer vertical resolution at the surface and increasing layer thickness at depth. The soil nodes of 0-1m are distributed at 0.01, 0.03, 0.08, 0.13, 0.23 ,0.33, 0.45, 0.55, 0.70, 1.05 m. Multiple snow layers are used to simulate the snow insulation effects accounting for changes in snow density and thermal properties due to seasonal snow cover evolution. The snow thermal properties including heat capacity and thermal conductivity are empirically estimated from snow density (Calonne et al., 2011). External model inputs include the upper boundary temperature conditions, total soil moisture content, and snow depth and density. Model outputs include soil temperature and unfrozen liquid water fraction along the soil profile, which is also used to define the soil F/T state.

The following 1-D heat transfer equation with phase change was used to simulate the snow and ground thermal dynamics:

$$C\frac{\partial}{\partial t}T(z,t) + L\zeta\frac{\partial}{\partial t}\theta(T,z) = \frac{\partial}{\partial z}\left(\lambda\frac{\partial}{\partial z}T(z,t)\right),$$

$$z \in [z_s, z_b]$$

(S1)

where $T(z,t)$ is the temperature (°C) at a specific soil depth ($z$) and time step ($t$), $L$ is the latent heat of fusion of water (J m$^{-3}$), $\zeta$ is the total soil water content (m$^3$ m$^{-3}$), and $\theta$ is the unfrozen liquid water fraction (%). $C$ and $\lambda$ are the volumetric heat capacity (J m$^{-3}$ K$^{-1}$) and thermal conductivity (W m$^{-1}$ K$^{-1}$) of soil respectively, varying with depth, soil moisture and F/T state. The upper boundary condition is set as the surface temperature at the snow/ground surface ($z_s$), while a heat flux characterizing the geothermal gradient is applied at the lower boundary ($z_b = 60\ m$). The soil thermal properties including the soil heat capacity and thermal conductivity are function of thermal properties of mineral and organic soil solid and liquid water, and ice components, weighted by their volumetric fraction.

The thermal conductivity $\lambda$ is estimated as a normalized thermal conductivity of the dry ($\lambda_{dry}$) and saturated ($\lambda_{sat}$) soil thermal conductivity weighted by soil saturation:

$$\lambda = K_e\lambda_{sat} + (1 - K_e)\lambda_{dry}$$

(S2)

where the Kersten number ($K_e$) is a function of the soil saturation degree, which uses a logarithm form for unfrozen soils and linear form for frozen soils (Farouki, 1981; Lawrence and Slate, 2008). $\lambda_{dry}$ is estimated from the soil bulk density. $\lambda_{sat}$ is estimated as a geometric mean of the thermal conductivity of different soil components (Farouki 1981), including mineral and organic soil solid, liquid water and ice, which can vary several-fold from highly organic soil (~0.5 W m$^{-1}$ K$^{-1}$) to mineral soils (1.5 ~ 3 W m$^{-1}$ K$^{-1}$).

Soil water usually freezes at a sub-zero temperature depending on solute concentration and other factors, and the model uses the following empirical function to estimate the unfrozen liquid water fraction ($\theta$):

$$\theta = \begin{cases} 1 & T \geq T_* \\ |T_*|^b \, |T|^{-b} & T < T_* \end{cases} \tag{S3}$$

The constant $T_*$ represents the freezing point depression, with values generally above -1°C (Woo, 2012). $b$ is a dimensionless parameter determined by fitting the unfrozen water curve, which can vary significantly depending on soil type (Schaefer and Jafarov, 2016).

**References:**

Calonne, N., Flin, F., Morin, S., Lesaffre, B., du Roscoat, S. R. and Geindreau, C.: Numerical and experimental investigations of the effective thermal conductivity of snow, Geophysical Research Letters, 38, L23501, doi:10.1029/2011GL049234, 2011.

Farouki, O.T.: Thermal properties of soils, Report No. 81(1), CRREL Monograph, 1981.

Lawrence, D. M. and Slater, A. G.: Incorporating organic soil into a global climate model, Climate Dynamics, 30, 145-160, 2008.

Rawlins, M. A., Nicolsky, D. J., McDonald, K. C., and Romanovsky, V. E.: Simulating soil freeze/thaw dynamics with an improved pan-Arctic water balance model, Journal of Advances in Modeling Earth Systems, 5, 659-675, 2013.

Schaefer, K. and Jafarov, E.: A parameterization of respiration in frozen soils based on substrate availability, Biogeosciences, 13(7), 1991–2001, doi:10.5194/bg-13-1991-2016, 2016.

Woo, M. K.: Permafrost hydrology, Heidelberg, Germany: Springer-Verlag, 575 pp, 2012.

Yi, Y., Kimball, J. S., Chen, R. H., Moghaddam, M., Reichle, R. H., Mishra, U., Zona, D. and Oechel, W. C.: Characterizing permafrost active layer dynamics and sensitivity to landscape spatial heterogeneity in Alaska, The Cryosphere, 12(1), 145–161, doi:10.5194/tc-12-145-2018, 2018.

Yi, Y., Kimball, J. S., Chen, R. H., Moghaddam, M. and Miller, C. E.: Sensitivity of active-layer freezing process to snow cover in Arctic Alaska, The Cryosphere, 13(1), 197–218, doi:10.5194/tc-13-197-2019, 2019.

---

## Author Response (AR1)

**Response to referee's comments on "Investigating the sensitivity of soil respiration to recent snow cover changes in Alaska using a satellite-based permafrost carbon model"**

**Dear Editor,**

We appreciate the constructive comments from the two reviewers, and have carefully revised the paper based on those comments. We added a flowchart and a brief model description to make the paper easier to follow; we also performed attribution analysis on the annual carbon fluxes to provide additional support to our conclusion. We also paid particular attention on the definition of soil respiration and removed "redundant" discussion throughout the results and discussion section.

Our responses to the comments are provided in the following text, and the revised manuscript is enclosed as a supplement with changes highlighted.

Thank you very much for considering our manuscript.

**Yonghong Yi**, on behalf of all authors

**Review 1#:**

1) General comments: "There are some problems. The results are wordy and fairly long; one problem is that there's a certain amount of discussion material mixed in. I suggest looking for opportunities to condense and cleanly separate different sections' material. I also was quite confused how you're comparing model output Rh with the Natali dataset, which is soil surface (Ra+Rh); there's a general carelessness with terminology in this area, confusing the reader about whether soil surface CO2 flux (soil respiration) or its heterotrophic component is being referred to. Finally, it's not acceptable, in my view, not to make the model code available at the review stage. For all these see below.
In summary, this is overall a strong, interesting, and well-done study. It would benefit from moderate revisions for clarity and concision in many places, and transparency and reproducibility absolutely need to be improved."

**Response:**

Thank you for the comments. We have carefully gone through the Results session (Section 3), and removed redundant discussion materials in Section 3 or moved them into Section 4, in order to be more concise, including section 3.1.1, 3.1.2, 3.2.2. Please refer to the manuscript for more details. We also redefined the "soil respiration" in the paper, and made it clear how we compared the model simulations with Natali's dataset. Please see the details in our response below. Finally, the code now is made public on GitHub: https://github.com/yiyh05/STM-C.

2) Line 30: "soil respiration" or heterotrophic respiration? I assume we're still talking about the latter, but clarify. Similarly line 31 mentions "total soil carbon emissions" – is this the Ra+Rh flux at the surface?

**Response:**

Here we are talking about "soil heterotrophic respiration". We revised the abstract and the title to be more specific.

3) L. 71: define soil respiration precisely here

**Response:**

We revised the text to clarify this:

Line 72-76: "Soil respiration is mainly the product of respiration by roots (autotrophic) and soil decomposers (heterotrophic), while it is generally difficult to partition soil respiration into the heterotrophic and autotrophic components (Phillips et al., 2017). In this study, we focus on the heterotrophic component of soil respiration, and assume it is the dominant component of total soil respiration in northern ecosystems during the cold season due to root dormancy (Tucker et al., 2014; Hicks Pries et al., 2015)."

4) L. 107: how are these depths chosen?

**Response:**

The model soil depth definition follows the previous model setup (Rawlins et al., 2013; Yi et al., 2015), with fine resolution at the soil surface and increasing thickness along the soil depth profile. The depth setup can change as long as there is finer vertical resolution at the surface, in order to ensure stability in solving the partial differential equations using finite-difference numerical methods, including both the 1-D soil heat transfer equation (Eq. S1) and soil carbon transport equation (Eq. 3). Please note that we have now moved this sentence to the model description in the supplementary material (S1).

5) L. 125: "linear"?

**Response:**

Yes, it is now corrected.

6) L. 129: interesting assumption. What's the rationale? Does litterfall = 100% of NPP in other systems, or at regional research sites?

**Response:**

Disturbances can significantly alter the balance between annual NPP and litterfall, which may be a large uncertainty source to our simulated NEE and respiration fluxes. In the northern high latitudes, fire disturbance is likely the largest contributing factor to this uncertainty. In other ecosystems, land use change and other disturbance events, such as harvest, insect damage, etc, can also be important depending on the region. However, modelling disturbance effects on the carbon balance at regional scale is generally a challenge. Also, please note that litterfall in this study also includes carbon turnover from woody components with a low turnover rate. The woody fraction from different land cover types can vary from 10% to 40% (Table S1), which was generalized from parameters used in the BIOME-BGC model and data collected in White et al. (2000).

7) L. 265: "therefore: : :" this logic is unclear. How the 2001-2016 period related to first part of sentence?

**Response:**

We clarified this in the main text:

Line 278-281: "Unfrozen conditions in the deep active layer may persist well into the cold season and even into January, causing a temporal lag in soil freeze onset at these depths that may extend into the following calendar year. Since the model was only run from 2001 to 2017, the soil freeze onset delay in year 2017 was not calculated."

8) L. 301-303: this sentence seems out of place

**Response:**

The differences in model simulated GPP at the US-Ivo site from the different plant functional types ("Shrub" vs "Tundra", Fig. 4) stem from the different maximum light use efficiency ($\varepsilon_{max}$) values prescribed for the two plant function types in the model parameterization (Table S1). The actual $\varepsilon_{max}$ can show large variability both across and within plant function types (Madani et al., 2014), which can contribute to model uncertainty. However, this information belongs in the discussion section, and so we removed this sentence to be more concise.

9) L. 304: perhaps start new paragraph here

**Response:**

We broke the paragraph into two as suggested by the reviewer due to a very long paragraph. Please refer to Line 334-345 for details.

10) L. 306-312: seems like discussion, not results

**Response:**

This was indeed discussion on what caused the mismatch between model and simulated carbon fluxes. However, the uncertainties in the MODIS LST, and the eddy covariance tower NEE partition method (mainly occur during the growing season) discussed here were not the focus of the discussion section. Therefore, we think it is better to place them here. However, we revised these sentences to be more concise:

Line 338-341: "The largest GPP reductions during the peak season were generally caused by very low nighttime LST, which may have large uncertainties in cloudy sky conditions. In addition, there is also large uncertainty imposed from the NEE partitioning method, with different methods resulting in large differences (up to more than1 g C m$^{-2}$ d$^{-1}$) in the tower-based GPP and R$_{eco}$ estimates."

11) L. 345: I'm confused how you're comparing model output Rh with the Natali dataset, which is soil surface (Ra+Rh)

**Response:**

We acknowledged the differences between Rh and the winter soil $CO_2$ flux from the Natali dataset, which includes both soil autotrophic and heterotrophic respiration. It would be more consistent if we directly compared the model simulated and measured Rh fluxes. However, there are very limited studies that provided reliable general partitioning methods between the autotrophic and heterotrophic respiration components of soil respiration for northern ecosystems.

On the other hand, the comparison was made only during the cold season defined from October from April. During this period, model simulated GPP is generally very low (especially for tundra); therefore, the model simulated total respiration is close to Rh. Also, some studies have

shown that Rh is the dominant component of total soil respiration in alpine and northern ecosystems during the cold season (Du et al., 2013; Tucker et al., 2014; Hicks Pries et al., 2015). Finally, our comparison focuses on the temperature sensitivity of carbon fluxes during the cold season between the model and the in-situ data.

We added a sentence in the methods to clarify this:

Line 238-241: "In this study, we compared the model simulated soil heterotrophic respiration directly with the measured soil $CO_2$ flux, since the model assumes the autotrophic respiration (as a portion of GPP) is very low throughout the cold season, especially for tundra (Tucker et al., 2014; Hicks Pries et al., 2015)."

12) L. 522-531: this seems unnecessary and duplicative of conclusions below

**Response:**

We removed this paragraph for brevity as suggested.

13) L. 560: perhaps start new paragraph for readability

**Response:**

We separated the paragraph into two as suggested by the review. Please refer to Line 587-589 for details.

14) L. 606-608: it's really inexcusable, in my view, to promise to upload data and code in the future while not making it available at the review stage

**Response:**

The code now is made public on GitHub: https://github.com/yiyh05/STM-C. The data produced by this study will be submitted to the ORNL DAAC as part of the NASA ABoVE archive; however, please note that it can take up to several months for the DAAC to review, accept and publish the resulting dataset.

**References:**

Du, E., Zhou, Z., Li, P., Jiang, L., Hu, X. and Fang, J.: Winter soil respiration during soil-freezing process in a boreal forest in Northeast China, Journal of Plant Ecology, 6(5), 349–357, doi:10.1093/jpe/rtt012, 2013.

Madani, N., Kimball, J. S., Affleck, D. L. R., Kattge, J., Graham, J., van Bodegom, P. M., Reich, P. B. and Running, S. W.: Improving ecosystem productivity modeling through spatially explicit estimation of optimal light use efficiency: Ecosystem optimal light use efficiency, J. Geophys. Res. Biogeosci., 119(9), 1755–1769, doi:10.1002/2014JG002709, 2014.

White, M. A., Thornton, P. E., Running, S. W., and Nemani, R. R.: Parameterization and Sensitivity Analysis of the BIOME–BGC Terrestrial Ecosystem Model: Net Primary Production Controls, Earth Interact., 4, 1–85, 2000.

Rawlins, M. A., Nicolsky, D. J., McDonald, K. C. and Romanovsky, V. E.: Simulating soil freeze/thaw dynamics with an improved pan-Arctic water balance model: SOIL FREEZE/THAW MODELING, J. Adv. Model. Earth Syst., 5(4), 659–675, doi:10.1002/jame.20045, 2013.

Yi, Y., Kimball, J. S., Rawlins, M. A., Moghaddam, M. and Euskirchen, E. S.: The role of snow cover affecting boreal-arctic soil freeze–thaw and carbon dynamics, Biogeosciences, 12(19), 5811–5829, doi:10.5194/bg-12-5811-2015, 2015.

Phillips, C. L., Bond-Lamberty, B., Desai, A. R., Lavoie, M., Risk, D., Tang, J., Todd-Brown, K. and Vargas, R.: The value of soil respiration measurements for interpreting and modeling terrestrial carbon cycling, Plant Soil, 413(1–2), 1–25, doi:10.1007/s11104-016-3084-x, 2017.

Hicks Pries, C. E., van Logtestijn, R. S. P., Schuur, E. A. G., Natali, S. M., Cornelissen, J. H. C., Aerts, R. and Dorrepaal, E.: Decadal warming causes a consistent and persistent shift from heterotrophic to autotrophic respiration in contrasting permafrost ecosystems, Glob Change Biol, 21(12), 4508–4519, doi:10.1111/gcb.13032, 2015.

Tucker, C. L., Young, J. M., Williams, D. G. and Ogle, K.: Process-based isotope partitioning of winter soil respiration in a subalpine ecosystem reveals importance of rhizospheric respiration, Biogeochemistry, 121(2), 389–408, doi:10.1007/s10533-014-0008-9, 2014.

**Review 2#:**

1) General comments: "In this study, Yi et al used a satellite-based permafrost carbon model to analyze the response of soil respiration to changes in snow coverage and temperature in the Alaska ecosystems. They concluded that for the time period from 2001 to 2017, soil respiration has overall increased with the warming. While I can sense the study was well attempted and carefully written, I feel some additional analyses may further improve the quantitative strength of some of the currently too colloquial conclusions. For instance, a time series plot showing how the carbon fluxes over Alaska have changed through the whole time period will give readers a more direct visual impression. In addition, in the trend analysis presented in Fig. 7, it is unclear how such trend change should be put into the context of changes in snow cover and warming. Perhaps an attribution analysis of these carbon flux trends to changes in snow cover, temperature, ALT, etc. will be helpful? Finally, maybe the authors could think of beginning the paper with a diagram (or flow chart) of how soil respiration is related to the variables they are investigating in this study? Such a diagram will put the attribution analysis (if the authors decide to do it) or the analysis in the result section into a better mental perspective."

**Response**:

Thank you for the suggestion. We have: 1) added a time series plot (Fig. S10), and performed additional attribution analysis (Fig. 9) to support our conclusion; and 2) added a diagram of the modelling framework (Fig. 1). We also added a description of the permafrost soil model that we used and also a few figures in the supplementary materials to address the reviewer's concerns. Please see our response below for more details.

2) Model description: the hydrological module is not well described. It took me quite a while to figure out the soil moisture is not simulated but rather is model input (am I right?). Moreover, in order to understand the model, I also read a number of other papers about the model, but was never clear how the whole model was assembled. So, if I may request, can the authors present a model description as supplemental material? Or at least give a list of what major variables are simulated, and what are prescribed as input.

**Response**:

The permafrost soil model (RS-PM) does not simulate the soil water movement directly; rather it uses the total soil water content from SMAP L4SM product as inputs to the model, which then simulates the soil freeze/thaw and associated changes in the unfrozen liquid water fraction. We have added a short paragraph in the beginning of Section 2.1 to more clearly illustrate the modeling process and the link between the permafrost soil model and the carbon model:

Line 103-112: "The Remote Sensing driven Permafrost Model (RS-PM) developed in Yi et al (2018; 2019), was coupled with a terrestrial carbon flux (TCF) model (Yi et al., 2015) to investigate the climate sensitivity of carbon fluxes across Alaska (Fig. 1), with a particular focus on the shoulder season. The soil decomposition model in the original TCF model was revised in this study to account for vertical soil carbon transport in order to better simulate the depth-dependent soil carbon distribution and respiration fluxes. The RS-PM model simulates soil temperature and changes in soil liquid water content due to soil freeze/thaw transitions along the

soil profile, using remote sensing based land surface temperature (LST), snow cover information and total soil moisture content as key model forcing. The RS-PM outputs were then used as inputs to the carbon model, as constraints on both the vegetation productivity and soil respiration. A brief description of the modeling framework is described here, with a focus on the revised soil decomposition model, while a detailed description on the RS-PM model is provided in the supplementary material."

The following flow diagram is presented in the new Figure 1. We also added more details on the permafrost soil model in the supplementary materials. Please refer to the supplement materials for more details.

[Figure]

**Fig. 1** Flow diagram describing the modelling procedure and main input datasets used in this study. The terrestrial carbon flux model has two components, including the light use efficiency algorithm for vegetation productivity estimates and a soil decomposition model for soil heterotrophic respiration estimates. The main equations used for each model component are indicated in the respective model boxes.

3) Fig 2, it is not easy to compare model with observations, even though I can see the model ballpark agrees with the response curve derived in Slate et al. (2017). The authors may consider interpolate the model results to the observations and present a scatter-plot as an addition to help analyzing the model performance.

**Response**:

There is a large discrepancy between downscaled MERRA2 (1-km) and in-situ effective snow depth data at the Snotel sites (Fig. S1a), so we chose not to directly compare the model simulated and in-situ soil temperature data at the Snotel sites. However, we do see overall consistency between the model simulated and in-situ soil temperature measurements at the 20cm reference depth as shown in Fig. S1b. Soil temperature data at 5 and 50 cm also show similar performance. We chose to include this figure in the supplementary material, rather than in the main text, to make the paper more concise.

[Figure]

**Fig. S1** Comparison between effective snow depth (a) derived from in-situ observations at Snotel sites and downscaled MERRA2 data, and observed and model simulated monthly soil temperature at 20 cm depth (b). Note that the sites compared for snow depth and soil temperature may be inconsistent due to inconsistency in the snow depth and soil temperature measurements at the Snotel sites. Generally, there are more snow depth measurements than soil temperature measurements.

4) Fig 3, it will be helpful to present a scatter-plot of modeled vs measured NEE.

**Response**:

The scatter-plots between modeled and measured NEE fluxes are now added as panel (d) in Fig. 4 (the original Fig. 3). The temperature sensitivity of ecosystem respiration at US-Atq in the original panel (d) is now presented as Fig. S2.

[Figure]

**Fig. 4** Model simulated carbon fluxes and temperature sensitivity of ecosystem respiration at two tundra sites (US-Ivo and US-Atq). "GPP1 obs" and "GPP2 obs" represent GPP estimates derived using tower-based NEE measurements and different partitioning methods provided by the tower PI, similar to "Reco1 obs" and "Reco2 obs". At the US-Ivo site, two GPP simulations were conducted using different maximum LUE parameters representing two different vegetation types (shrub and grassland tundra), indicated as "GPP (shrub)", and "GPP (tundra)" in panel (a). Comparisons between model and tower-based NEE fluxes at the two sites are shown in panel (d).

5) Fig 6. Panel c and d are hard to compare, maybe the authors can consider contrasting two depths each panel in two panels, so readers can compare the time series more straightforwardly.

**Response**:

According to the reviewer's suggestion, we now combined panel (c) and (d) in Fig. 7 (originally as Fig. 6) as a single panel, and compared the depth-dependent Rh fraction for the two permafrost zones. We combined the two intermediate soil depths (13-33 cm, 33-55 cm) as a single depth (13-55 cm), to be more concise.

[Figure]

**Fig. 7** Regional mean of model simulated carbon fluxes (a), Rh fluxes from different soil depths (b) averaged across Alaska, and Rh contribution from different soil depths to total Rh averaged across two regions with different permafrost probability (c). In panel (c), solid and dashed lines represent the mean values averaged across areas with permafrost probability from 0-33% and 67-100%, respectively. Gray shading denotes the standard deviation of monthly mean fluxes from 2001 to 2017.

6) Fig. 7, like in my major comments, if a quantitative attribution analysis can be done here, it will be very helpful.

**Response**:

We added two figures to support quantitative analysis for the results as requested by the reviewer: 1) Fig. S10 shows the time series plot of the annual carbon fluxes; 2) Fig. 9 shows the relative importance of selected climate variables to the annual carbon fluxes. The original Fig. 9 that provides results on the correlation analysis between Rh fraction and seasonal LST is now moved to the supplementary material (Fig. S13) to make the paper more concise.

The attribution analysis was conducted using the gradient boosting regression method, and is described in Section 2.4 (Line 288-304):

"Finally, we used the gradient boosting regression (GBR) method to quantify the contribution of selected environmental variables to the annual carbon fluxes. The GBR method consists of a sequence of models, and each consecutive model is developed based on the errors of previously added models (Friedman, 2000). The above model simulated annual carbon fluxes from 2002 to 2017 were used to train and evaluate the GBR models. We chose the following nine contributing environmental factors or predictors to annual carbon fluxes during the model fitting, including summer (June-August) NDVI, annual freezing and thawing index, mean annual downward solar radiation, rootzone soil moisture during the thaw season, snow offset and onset, mean snow depth averaged from January to March (representing annual maximum snow depth), and snow depth during the early snow season (from October to November). The GBR method was implemented using the sklearn package in Python 2.7. The following method was used to determine the relative importance of each predictor to the model predictive performance. We first ran the model using all nine predictors, and the model results were referred as the baseline simulation ($GBR_{baseline}$). We then ran the fitted model successively with one randomized variable and the other variables intact, with the model outputs denoted as $GBR_{one\_variable\_randomized}$. The reduction in the Pearson's correlation coefficient between the two model runs was used to quantify the relative importance of each variable, computed as follows (Karjalainen et al., 2019; Zheng et al., 2020):

$$I_x = 1 - corr(GBR_{baseline} - GBR_{one\_variable\_randomized})$$
$$RI_x = \frac{I_x}{\sum_{x=1,9} I_x} \tag{6}$$

where $I_x$ represents the reduction in the correlation coefficient of the model runs with the variable $x$ randomized, and $RI_x$ is the relative importance value of variable $x$."

These results were added in section 3.2.1 (Line 415-431) of the revised paper:

"…At the regional scale, the time series of estimated annual carbon fluxes showed non-significant (p > 0.1) positive trends of 2.58, 1.86, and 0.38 Tg C yr[-1] for respective GPP, Rh and NEE fluxes (Fig. S10).

The attribution analysis results using the GBR method confirmed that NDVI and annual thawing index are the two most important variables affecting the estimated annual carbon fluxes, which was generally consistent across different vegetation types (Fig. 9). For annual GPP flux, NDVI was the most important variable followed by annual thawing index and downward solar radiation, while for annual Rh fluxes, annual thawing index was the most important variable, followed by NDVI, with other variables playing a very minor role. Despite the importance of annual thawing index in controlling annual GPP and Rh fluxes, the snow offset showed little importance to both fluxes. This was likely due to the low temporal resolution of the MODIS snow cover data (i.e. 8-day composite) used to calculate the snow offset, which was calculated as the center date of the 8-day composite period. The low temporal resolution of snow offset and a strong correlation (R>0.7, p<0.1) between annual thawing index and snow offset may limit its use in the regression model. As for annual NEE flux, NDVI, downward solar radiation, and annual freezing index are among the most important factors. However, the effects of different variables on annual NEE

flux varied throughout the period due to their compensating effects on GPP and Rh fluxes, and NEE being a small residual of these two larger carbon fluxes; therefore, none of the variables played a dominant role throughout the entire period. In addition, the GBR model also showed generally poor performance in predicting annual NEE fluxes (R ≥ 0.7) compared with the other two fluxes (R > 0.9)."

[Figure]

**Fig. 9** Mean relative importance values of selected environmental variables in controlling model estimated annual carbon fluxes in Alaska (a: GPP; b: Rh; c: NEE). The importance values were averaged for four major vegetation types (Forest, Shrub, Herbaceous, and Wetlands, Fig. 2), and the error bar represents their standard deviation across the different vegetation types. The nine environmental variables are: summer (June-August) NDVI, annual thawing and freezing index, snow offset and onset, mean snow depth averaged from January to March (representing annual maximum snow depth), and snow depth averaged during the early snow season (from October to November), mean annual downward solar radiation, and rootzone soil moisture during the thaw season. The annual thawing and freezing index represent the sum of MODIS LST above 0 °C and below 0 °C throughout the year, respectively.

[Figure]

**Fig. S10** Time series of annual carbon fluxes summed over the Alaska study area (~1.21 million km$^2$) from 2001 to 2017. Gray shading denotes the standard deviation of estimated annual NEE flux over the study area. A very low standard deviation of NEE flux in 2001 was due to the model steady state

assumption in the spin up year (2001). The standard deviation of GPP and Rh flux across the study area was approximately 50% of the regional mean, and was not shown.

7) Other minor comments: L 204 "soil moisture" is unclear, maybe "liquid water" should be used.

**Response**:

We now use "liquid water content" instead of "soil moisture".